# ShapeEmbed: a self-supervised learning framework for 2D contour quantification

Anna Foix Romero[1], Craig Russell[1], Alexander Krull[*2], and Virginie Uhlmann[*1,3]

[1]European Bioinformatics Institute, European Molecular Biology Laboratory, Cambridge, UK,
`{afoix, ctr26, uhlmann}@ebi.ac.uk`
[2]School of Computer Science, University of Birmingham, Birmingham, UK,
`a.f.f.krull@bham.ac.uk`
[3]Department of Molecular Life Sciences, University of Zurich, Zurich, CH,
`virginie.uhlmann@mls.uzh.ch`

[*]Corresponding authors

## Abstract

The shape of objects is an important source of visual information in a wide range of applications. One of the core challenges of shape quantification is to ensure that the extracted measurements remain invariant to transformations that preserve an object's intrinsic geometry, such as changing its size, orientation, and position in the image. In this work, we introduce ShapeEmbed, a self-supervised representation learning framework designed to encode the contour of objects in 2D images, represented as a Euclidean distance matrix, into a shape descriptor that is invariant to translation, scaling, rotation, reflection, and point indexing. Our approach overcomes the limitations of traditional shape descriptors while improving upon existing state-of-the-art autoencoder-based approaches. We demonstrate that the descriptors learned by our framework outperform their competitors in shape classification tasks on natural and biological images. We envision our approach to be of particular relevance to biological imaging applications.

## 1 Introduction

The outline of objects in 2D images carries essential information about their shape. In natural images, humans are often able to recognize objects purely based on their silhouette without relying on texture or color [Wagemans et al., 2008]. Shape information is unaltered by many geometric operations such as similarity transformations [Dryden and Mardia, 2016] and is also unaffected by irrelevant and distracting imaging variables, such as lighting conditions or imaging setups. This is particularly relevant in biological imaging, where the shapes of living systems extracted from microscopy images serve as phenotypic fingerprints to reveal cell identity, cell states, and response to chemical treatments across a wide range of imaging scales, settings, and modalities [Paluch and Heisenberg, 2009, Rangamani et al., 2013, Grosser et al., 2021, Zinchenko et al., 2023]. All of these aspects make shape a highly desirable abstraction from pixel-intensity based images, enabling visualization, outlier detection, and unsupervised discovery of underlying patterns [Loo et al., 2007, Sailem et al., 2015].

The standard way of describing objects in 2D images is with binary segmentation masks, where pixels inside of an object's outline are set to $1$ and pixels outside to $0$. However, while such a representation is readily produced by segmentation algorithms and allows for abstracting from lighting and imaging conditions, it is not invariant to transformations such as translation, rotation, reflection, and scaling. As such, the same object appearing twice in an image at a different location or orientation will yield segmentation masks that can only be recognized as equivalent after tedious processing.

39th Conference on Neural Information Processing Systems (NeurIPS 2025).

To circumvent this and preserve invariance to similarity transformations, shape information is traditionally captured through statistics computed from the mask image, such as region properties (*e.g.*, area and curvature) or Fourier descriptors [Pincus and Theriot, 2007]. Such methods, however, are averaging and condensing information by design, thus providing an incomplete description from which it is impossible to fully reconstruct the original outline in all of its details.

Representation learning has recently gained attention as a strategy utilizing autoencoders [Hinton and Salakhutdinov, 2006, Kingma and Welling, 2014] to derive descriptors that can capture all intricacies of object shapes while producing descriptors that are invariant to irrelevant geometric transformations. The vast majority of the methods proposed so far [Chan et al., 2020, Ruan and Murphy, 2019, Vadgama et al., 2022, 2023] aim to encode segmentation masks by relying on complex training strategies to ensure that the resulting latent code representations are geometrically invariant.

Here, we introduce ShapeEmbed, a novel approach to extract shape descriptors relying on representation learning that leverages a simple architecture and training procedure to ensure invariance to translation, scaling, rotation, and reflection. Instead of directly encoding segmentation masks, we propose to encode instead a distance matrix [Dokmanic et al., 2015] representation of object outlines. The distance matrix contains all pairwise distances of the points on the outline of an object and is inherently invariant to translation and rotation. It also fully describes the object contour and allows reconstructing it via multi-dimensional scaling (MDS) [Cox and Cox, 2000] without loss of information. On the other hand, distance matrices are not invariant to indexation. Even for simply connected outlines described as a sequence of ordered points, the choice of the origin and direction of travel of the sequence will result in different distance matrices. These different distance matrices will, however, be equivalent up to elementary permutations of rows and columns. Leveraging this property, we can implement invariance to indexation in the encoding step through a specific architecture of the encoder and the inclusion of a new loss function, leading to a latent descriptor of shape that is robust to all shape-preserving geometric transformations.

Distance matrices have been used for a long time to characterize shapes and to compute shape dissimilarities without alignment [Hu et al., 2012, Konukoglu et al., 2012, Govek et al., 2023]. While the use of pairwise point distances in these previous works is similar to what we propose, we do not use the point distances directly but instead as an input to a representation learning model that maps outlines to points in a latent shape descriptor space with generative properties. Our approach has similarities with Alphafold [Jumper et al., 2021], where distance matrices are used to describe the structure of proteins. However, as proteins are open linear structures with a clearly defined start and end point, the problem of indexation invariance encountered with closed outlines does not arise. We are thus, to the best of our knowledge, the first to overcome this issue and propose a framework to encode distance matrices of closed 2D contours with a VAE.

We evaluate our method by using a simple logistic regression classifier applied to the latent representation as a downstream shape classification task. We demonstrate that ShapeEmbed outperforms traditional statistics-based as well as learning-based methods on a range of different problems, including computer vision benchmarks and biological imaging datasets. Further quantitative exploration of the structure of our latent space indicates that its structure captures meaningful aspects of the shape of objects in images. In summary, our main contributions are as follows:

1. We introduce, to the best of our knowledge, the first self-supervised representation learning model that learns shape descriptors from distance matrices. The descriptors are, by design, invariant to scaling, translation, rotation, reflection, and re-indexing.

2. We are, to the best of our knowledge, the first to propose a solution to achieve indexation invariance in a VAE architecture for shape description based on a padding operation in the encoder, operating jointly with a new loss function.

3. We show that our method outperforms the representation learning state-of-the-art and classical baselines on downstream shape classification tasks.

## 2 Related Work

We hereafter review relevant prior works on image-based shape quantification.

**Statistics-based methods.** Shape quantification relying on summary statistics aims to assemble a large enough collection of features, assuming that their ensemble provides a sufficiently complete

description of the object's shape. The features themselves are handcrafted by design and most often consist of quantities such as area, perimeter, and curvature [van der Walt et al., 2014]. Due to its simplicity and good empirical performance, this approach is overwhelmingly used in biological imaging [Bakal et al., 2007, Barker et al., 2022]. Many summary statistics are inherently invariant to geometric transformations such as rotation and translation, but only partially capture shape information. As such, they are often unable to distinguish subtle shape differences.

**Decomposition methods.** Decomposition methods seek to approximate an object's shape by a set of basis elements. The shape descriptor then corresponds to the coefficients of that approximation, and the original outline can be reconstructed as a weighted sum of the basis elements. The most common example of decomposition-based shape descriptors are the Elliptical Fourier Descriptors (EFD, [Persoon and Fu, 1977, Kuhl and Giardina, 1982]). EFD are inherently invariant to similarity transformations, but often perform poorly in classification tasks as discriminative information tends to be hidden in noisy higher-order approximation coefficients.

**Learning-based methods.** Following the success of autoencoders [Hinton and Salakhutdinov, 2006], variational autoencoders (VAE, [Kingma and Welling, 2014]), contrastive learning [Chen et al., 2020, He et al., 2020], and vision transformers [Caron et al., 2021, He et al., 2022] for representation learning, self-supervised learning of shape descriptors directly from object masks appeared as a natural strategy to alleviate the shortcomings of classical methods. Methods have been proposed to encode images of 2D objects into a latent representation of the underlying object's shape [Chan et al., 2020, Zaritsky et al., 2021], but are often not invariant to translation, scaling, and rotation. To mitigate this issue, a generic prealignment step can be carried out [Ruan and Murphy, 2019]. However, as shown in [Burgess et al., 2024], it does not consistently produce good results.

A framework that employs invariant risk minimization to learn invariant shape descriptors was recently introduced in [Hossain et al., 2024]. The approach focuses on capturing invariant features in latent shape spaces parameterized by deformable transformations. While being robust to environmental variations, this method does not explicitly focus on achieving invariance to geometric transformations in the resulting shape representations and is heavily tailored to medical imaging data, with limited applicability to other types of images.

The points composing the contour of a 2D mask can be viewed as a 3D point cloud where all points lie on a plane, thus making recent works towards rotation-invariant point networks potentially relevant to the problem of 2D shape representation learning. Several rotation invariant architectures [Li et al., 2021a, Zhang et al., 2022, Li et al., 2021b] have been proposed for classification, segmentation, and shape retrieval in a supervised fashion, but fewer consider the problem of learning shape representations without any labels [?Furuya et al., 2024]. Processing point clouds, which are by definition sets and therefore unordered, however, differs from processing contours, defined in our case as ordered sequences of points. Relying on ordered sequences, as we do in our approach, is critical to maintain information about point connectivity and straightforwardly reconstructing outlines for visualization, which is essential for biological imaging applications.

Recently, [Vadgama et al., 2022, 2023] introduced a VAE model trained to produce a latent space that explicitly disentangles a 2D geometric shape descriptor from the orientation of the input object. The decoder network takes the orientation-invariant shape descriptor together with the orientation as input and is thus able to reconstruct the original 2D contour. Both of these methods are superficially similar to ours in that they use a VAE and achieve rotation invariance. However, while [Vadgama et al., 2022, 2023] explicitly estimate a rotation using their encoder network, our method bypasses this step by using the already rotation-invariant distance matrix representation as input to the encoder. As neither of these works evaluates their method on a downstream task and unfortunately do not provide a code repository, we were unable to include them in our results comparison.

Most closely related to our work is O2VAE [Burgess et al., 2024], a VAE model that encodes segmentation masks into an orientation-invariant latent code representation. The key idea of this approach is to rely on an encoder with rotation-equivariant convolutional layers [Weiler and Cesa, 2019] together with pooling to achieve invariance. In the O2VAE model, a realignment step is required during training to orient the input with its reconstruction. While O2VAE uses an elaborate special encoder to achieve rotation invariance, our method is inherently rotation-invariant due to its use of a distance matrix representation and only requires simple modifications to the VAE architecture to achieve indexation invariance.

# 3 Proposed Approach

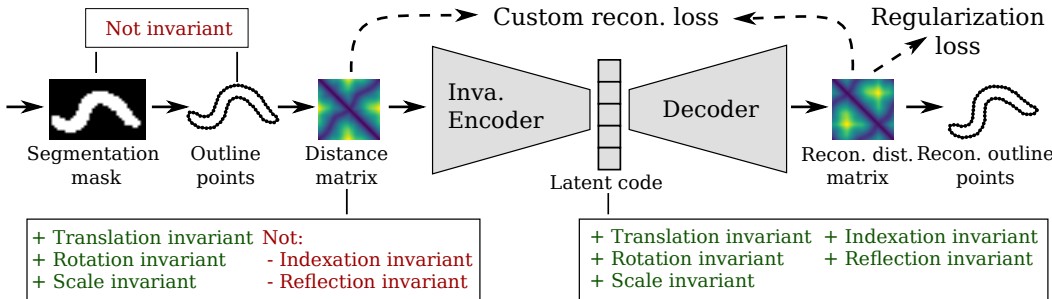

Figure 1: Overview of ShapeEmbed. ShapeEmbed converts the outline of an object from a 2D segmentation mask into a normalized distance matrix representation that is translation, rotation, and scale invariant. Relying on a VAE model, it then encodes distance matrices into a latent representation that adds indexation and reflection invariance. The resulting latent code forms a powerful shape descriptor that can be used for downstream tasks such as classification, and allows for reconstructing the original outline, albeit arbitrarily indexed, rotated, translated, and reflected.

ShapeEmbed extracts the outline of objects in 2D segmentation masks to construct a distance matrix representation that is then used to train a VAE model to learn a latent representation of shape. Thanks to a combination of the distance matrix properties and the VAE model design, the resulting latent codes are invariant to translation, rotation, reflection, scaling, and contour point indexation (Figure 1). In the following, we describe ShapeEmbed step-by-step and discuss how we achieve these different types of invariance in our framework.

## 3.1 From Segmentation Masks to Distance Matrices

ShapeEmbed operates with contours that are simply connected and described by an ordered sequence of successive points. Starting with a 2D binary segmentation mask, we first extract a simply-connected pixel outline. We use the marching squares algorithm [Lorense, 1987], but other methods would be equally applicable. We then interpolate the pixels of the outline with a parametric linear spline curve that we uniformly sample starting at an arbitrary position on the outline and going counterclockwise to yield a sequence of $N$ successive points $\mathbf{x}_i = (x_i, y_i)$. $N$ is a hyperparameter that we set to 64 by default, and that can be adjusted depending on the number of pixels composing the outline. We then construct the corresponding $N \times N$ distance matrix $D$ with entries $d_{i,j} = |\mathbf{x}_i - \mathbf{x}_j|$, which is the Euclidean distance between points $\mathbf{x}_i$ and $\mathbf{x}_j$. Distance matrices are naturally invariant to translation and rotation. To make them additionally invariant to scaling, we normalize them using the matrix norm, as formally demonstrated in Supplementary Section A.

Despite the sequence being ordered to capture the succession of points along the object contour, our distance matrices are sensitive to the choice of origin (starting point) in the sequence and direction of travel (clockwise or counterclockwise), both of which impact the ordering of the matrix entries. Upon changing the starting point and/or direction of travel, the matrix entries will be shifted diagonally (change of origin) as well as horizontally and vertically mirrored (change of direction of travel), as illustrated in Figure 2. More precisely, for a given distance matrix $D$, we denote the equivalent distance matrices obtained by choosing point number $k \in \{0, \ldots, N-1\}$ as origin and $o \in \{1, -1\}$ as direction of travel as $D^{k,o}$. This yields a total of $2N$ different equivalent matrices representing the same point sequence. The matrix entries are given by

$$d_{i,j}^{k,o} = d_{(io+k) \bmod N,\ (jo+k) \bmod N,} \tag{1}$$

where $d_{i,j}$ are the entries of the original distance matrix $D$.

We propose a minor modification to the encoder architecture in our VAE that makes it unable to distinguish between these re-indexations. Together with a modified loss function, our VAE is thus guaranteed to map all possible equivalent indexings of successive outline points to the same latent vector. Importantly, indexation invariance also grants our approach invariance to mirror reflection: assuming a fixed choice of origin and direction of travel, a mirror reflection of the outline will

indeed correspond to a change of direction of travel, resulting in a distance matrix that is mirrored horizontally and vertically.

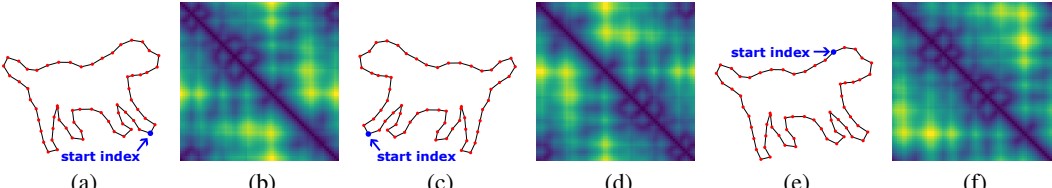

| (a) | (b) | (c) | (d) | (e) | (f) |

Figure 2: Effect of indexation changes on the distance matrix. A point outline (a) and its corresponding distance matrix (b) obtained by traveling the sequence of successive outline points counterclockwise from a given choice of origin (start index). Changing the direction of travel is equivalent to traveling through a mirror-reflected version of the outline in the counterclockwise direction (c) and yields a distance matrix that is mirrored horizontally and vertically (d). A different choice of origin (e) produces a diagonally-shifted version (f) of the original distance matrix (b).

## 3.2 VAE Model with Custom Indexation Invariant Encoder

ShapeEmbed relies on a VAE model that encodes distance matrices into a latent code representation that is invariant to shape-preserving transformations of the original outline.

Since distance matrices are 2D structures, they naturally lend themselves to being processed by powerful and established convolutional backbones developed for image data [Bengio et al., 2013]. In our implementation, we thus use an encoder network based on the ResNet-18 architecture [He et al., 2016] that we mirror in the decoder path.

Remembering that our normalized distance matrices are naturally invariant to translation, rotation, and scaling but not to point indexation, we designed a novel indexation invariant encoder architecture to ensure that our latent codes only carry information about intrinsic shape. As outlined in 3.1, different choices of origin in the outline point sequence result in distance matrices that are shifted diagonally. Conveniently, the convolutional layers in ResNet-18 are shift equivariant, meaning that a shifted input will result in an identical but shifted output. Carefully considering boundary conditions, we propose to use circular padding (*i.e.*, padding by repeated tiling) in every convolutional layer, which directly corresponds to the modulo operation in 1. As a result, the convolutional layers are shift-equivariant and produce equal but shifted outputs for all possible distance matrix indexations (starting points). By combining convolutions with a subsequent global pooling operation, we could in principle achieve true shift-invariance. However, ResNet-18 reduces the tensor size in multiple steps using local pooling operations and strided convolutions. Strictly speaking, when convolutions are used within such architectures, the result is therefore no longer truly shift equivariant or invariant [Rumberger et al., 2021]. In practice, we however observe that our architecture is sufficient to help prevent the latent codes from capturing indexation changes, as demonstrated experimentally in Section 4.

Our final encoder backbone is therefore a modified ResNet-18 where the standard convolutional and pooling operations have been replaced with layers that incorporate circular padding. To make our encoder additionally invariant to the direction of travel of the outline, we process each matrix twice using the backbone, once in its original form and once horizontally and vertically mirrored. We then sum the two resulting output vectors to create an architecture that is unable to distinguish between a matrix and its mirrored version, rendering it invariant to reflection.

## 3.3 Loss Function

**Indexation Invariant Reconstruction Loss.** Considering that our encoder is sufficiently invariant to indexation, it follows that next to no information about the indexation of the point sequence is present in the latent code. This is a problem when computing the reconstruction loss: as the same latent code could have been created by any shifted and mirrored version of the distance matrix, it is impossible to know which version of the matrix should be reconstructed to match the input - any of these alternative versions is correct as they all generate the same point sequence. To account for this ambiguity, we introduce a novel reconstruction loss that equally rewards all equivalent versions. To

compute it, we generate all $2N$ alternative versions $D^{k,o}$ of the input distance matrix. We then define the reconstruction loss as

$$\mathcal{L}_{\text{rec}}(\hat{D}, D) = \min_{k \in \{0,...,N-1\}, o \in \{-1,1\}} \text{MSE}(\hat{D}, D^{k,o}), \tag{2}$$

where $\hat{D}$ is the decoded distance matrix (reconstruction), $D$ is the true distance matrix (input), $D^{k,o}$ is an alternatively indexed version of $D$ (see (1)), and $\text{MSE}(\cdot, \cdot)$ is the mean squared error over all matrix entries. This approach ensures that the decoder learns to reconstruct a version of the input distance matrix that minimizes the reconstruction error regardless of indexation. It thus effectively removes the ambiguity without losing indexation invariance. By incorporating this loss into the training process, the model is encouraged to focus on the intrinsic geometric structure of the outlines rather than being sensitive to the arbitrary choice of indexation of the sequence of points composing them.

**Distance Matrix Regularization Losses.** We use several Euclidean distance matrix properties to regularize the learning process and encourage the decoder to produce a distance matrix-like output, leading to the formulation of three regularization terms.

First, as the distance from a point to itself is null, all entries in the leading diagonal of the distance matrix should be zero, which translates to $\mathcal{L}_{\text{diag}}(\hat{D}) = \frac{1}{N} \sum_{i=1}^{N} \hat{d}_{i,i}^2$, where $\hat{d}_{i,j}$ is the $i$th entry in the diagonal of $\hat{D}$. Secondly, as the Euclidean distance is non-negative, all entries should be greater than or equal to zero, which translates to $\mathcal{L}_{\text{non}-\text{neg}}(\hat{D}) = -\frac{1}{N^2} \sum_{i=1}^{N} \sum_{j=1}^{N} \min(\hat{d}_{i,j}, 0)$. Third and finally, since the Euclidean distance is symmetric, the matrix should be symmetric too, which translates to $\mathcal{L}_{\text{sym}}(\hat{D}) = \text{MSE}\left(\hat{D}, \hat{D}^{\top}\right)$.

**Overall Loss.** Putting everything together, we use the following weighted sum as a loss to train our model:

$$\mathcal{L}_{\text{VAE}} = \mathcal{L}_{\text{rec}} + \beta \mathcal{L}_{\text{KL}} + \gamma \mathcal{L}_{\text{diag}} + \delta \mathcal{L}_{\text{non}-\text{neg}} + \epsilon \mathcal{L}_{\text{sym}}, \tag{3}$$

where $\mathcal{L}_{\text{KL}}$ is the classical Kullback-Leibler divergence loss [Kingma and Welling, 2014], $\mathcal{L}_{\text{rec}}$ is our custom reconstruction loss (2), and $\beta$, $\gamma$, $\delta$, and $\epsilon$ are scalar hyperparameters. The hyperparameter $\beta$ allows tuning the model to focus more on feature extraction and reconstruction (smaller $\beta$) or on producing a smooth latent space that can be used in a generative context (larger $\beta$) [Higgins et al., 2017]. We empirically set it to $10^{-10}$ by default, as this value was observed to balance accurate reconstructions and meaningful sampling in the latent space. The hyperparameters $\gamma$, $\delta$, and $\epsilon$ are all set by default to $10^{-5}$, which was empirically found through hyperparameter tuning.

### 3.4 Outline Reconstruction

Although we assess the latent representation learned by ShapeEmbed in downstream shape quantification tasks, it is useful to be able to reconstruct outlines from the latent codes for visualisation and quality control purposes. Outline points can be retrieved from a distance matrix using the MDS algorithm [Cox and Cox, 2000]. However, despite the regularization terms presented in 3.3, the outputs of ShapeEmbed are neither truly symmetric nor have a leading diagonal composed of perfect zeros, and are therefore not true distance matrices. These deviations are fortunately typically negligible and within numerical error range, meaning that the leading diagonal values can be set to zero without significant loss of information. To enforce symmetry, we also take the average of the matrix and its transpose as $\frac{1}{2}(\hat{D} + \hat{D}^{\top})$. This operation is guaranteed to produce a symmetric matrix, thus allowing us to apply MDS. The algorithm is initialized with a random set of 2D points and iteratively updates them to minimize the difference between the entries of the distance matrix and the Euclidean distances between the points. MDS is guaranteed to converge, but not to the same solution every time.

However, the solutions it recovers are all equivalent up to rotation, translation, and reflection, meaning that the resulting outline will be arbitrarily rotated, translated, and reflected. Since the distance matrices inputted to the model are normalized for scale, the matrix norm must be carried over to the post-processing step and applied to the output distance matrix before MDS if one wants to recover the originally-sized outline.

# 4 Experiments

In this section, we review the datasets and evaluation metrics we use in our experiments, provide the implementation details of our method, present and discuss the performance of ShapeEmbed against relevant competitors, perform in-depth ablation studies to inspect the importance of the various invariance properties granted by our model, and finally demonstrate the added value of ShapeEmbed to identify subtle phenotypes in biological images. ShapeEmbed is implemented in Python and is available at `https://github.com/uhlmanngroup/ShapeEmbed` under the MIT license. Further implementation details are provided in Supplementary Section B.

## 4.1 Datasets

**MNIST.** The MNIST benchmark dataset (GNU GPL, [Deng, 2012]) consists of grayscale images of handwritten digits from 0 to 9, with approximately $7,000$ images per class, amounting to a total of $70,000$ images.

**MPEG-7.** The MPEG-7 CE-Shape-1 Part B dataset (LGPL-3.0, [mpe, 2009]) is a benchmark for shape matching and retrieval tasks. It consists of $1,400$ binary masks of objects belonging to 70 classes, with 20 images per class. Each class represents a distinct object category, such as different animals, tools, or symbols, designed to cover a range of shape variability.

**BBBC010.** The Broad Bioimage Benchmark Collection 10 (BBBC010, no license, [Ljosa et al., 2012]) is a biological imaging dataset designed to test phenotypic profiling at the whole-organism level. It contains a total of $1,407$ individual binary masks of *C. elegans* nematodes divided into a live and a dead class, each containing 768 and 639 individuals, respectively.

**MEF.** The Mouse Embryonic Fibroblast (MEF, MIT License, [Phillip et al., 2021]) dataset is a challenging biological imaging dataset containing 300 images of multiple cells distributed across three classes: circle-patterned, triangle-patterned, and control (non-patterned) surfaces, with 100 images per class. Although the original dataset includes two color channels corresponding to an actin and a nuclei stain, we here only use the actin channel as it captures whole cells. Binary masks of individual cells are provided, leading to a total of $26,198$ objects distributed into $3,192$ in the control, $6,624$ in the triangle, and $6,565$ in the circle class, respectively.

**HeLa Kyoto.** The HeLa Kyoto dataset (CellCognition project, CC-BY 4.0 License, [Held et al., 2010]) consists of fluorescence microscopy images of H2B-mCherry-stained HeLa Kyoto cell nuclei, labeling chromatin and capturing nuclear morphology during mitosis. Already segmented and cropped masks for individual nuclei are available under the "classifier data" section of the dataset. We consider 313 objects in 4 classes that are representative of nuclei at key phases of mitosis (category name and number of samples in parentheses): early anaphase (earlyana, 40), lateana (lateana, 83), metaphase (meta, 110), and prometaphase (prometa, 80).

**Mouse Osteosarcoma Cells (MOC).** The MOC dataset (MIT license, [Miolane et al., 2020]) consists of fluorescence microscopy images of mouse osteosarcoma cells. In this dataset, cells have been exposed to cytoskeletal perturbation through treatment with the single drugs jasplakinolide (Jasp) and cytochalasin D (Cytd). The 649 cells are divided in 3 classes: a control class with 318 untreated cells, a Cytd class with 175 cells, and a Jasp class with 156 cells.

Additional details on experimental settings for each datasets, as well as example images and masks illustrating the two bioimaging datasets considered are provided in Supplementary Section C.

## 4.2 Baselines and Evaluation Strategy

We compare the performance of ShapeEmbed for shape classification against two classical shape analysis baselines (Elliptical Fourier Descriptors [Persoon and Fu, 1977] and Region Properties [van der Walt et al., 2014]), against two state-of-the-art representation learning models (the contrastive learning framework SimCLR [Chen et al., 2020] and the vision transformer Masked Autoencoder [He et al., 2022]), and against our direct competitor (O2VAE [Burgess et al., 2024]). Additional details on the implementation of these methods are provided in Supplementary Section D.

To quantitatively evaluate the quality of the different shape descriptors we consider, we rely on a downstream classification task. We train a logistic regression classifier [Bisong, 2019] following

Table 1: Benchmark datasets results (F1-score $\pm \sigma$, higher is better).

| METHOD | MNIST | MPEG-7 |
|---|---|---|
| REGION PROP. | $0.81 \pm 0.00$ | $0.70 \pm 0.01$ |
| EFD | $0.62 \pm 0.01$ | $0.08 \pm 0.01$ |
| SIMCLR | $0.59 \pm 0.01$ | $0.13 \pm 0.02$ |
| MAE (VIT-B) | $0.95 \pm 0.03$ | $0.65 \pm 0.00$ |
| MAE (VIT-L) | $0.84 \pm 0.01$ | $0.63 \pm 0.04$ |
| MAE (VIT-H) | $0.85 \pm 0.01$ | $0.60 \pm 0.01$ |
| MAE (VIT-B, PRETRAINED) | $0.92 \pm 0.01$ | $0.42 \pm 0.01$ |
| MAE (VIT-L, PRETRAINED) | $0.93 \pm 0.01$ | $0.51 \pm 0.02$ |
| MAE (VIT-H, PRETRAINED) | $0.94 \pm 0.01$ | $0.57 \pm 0.02$ |
| O2VAE | $0.86 \pm 0.01$ | $0.13 \pm 0.02$ |
| **SHAPEEMBED** | $\mathbf{0.96 \pm 0.01}$ | $\mathbf{0.75 \pm 0.02}$ |

Table 2: Scaling and indexation invariance ablation results (F1-score $\pm \sigma$, higher is better). "None" indicates no indexation invariance and no scale normalization.

| METHOD | sMNIST | sMPEG |
|---|---|---|
| NONE | $0.87 \pm 0.01$ | $0.24 \pm 0.03$ |
| NO INDEX. INV. | $0.88 \pm 0.01$ | $0.59 \pm 0.07$ |
| NO NORM. | $0.91 \pm 0.01$ | $0.42 \pm 0.02$ |
| **SHAPEEMBED** | $\mathbf{0.95 \pm 0.00}$ | $\mathbf{0.70 \pm 0.09}$ |

a 5-fold cross-validation strategy, and report the mean and standard deviation of the F1-score as a performance metric over the 5 data folds. The F1-score balances precision and recall and thus provides a reliable measure of performance across the considered datasets [Ye et al., 2012], with a higher F1-score indicating better performance. For the benchmarking and biological imaging experiments, we also report additional metrics (log loss, accuracy, precision, and recall) in Supplementary sections.

### 4.3 Benchmarking

We quantitatively evaluate the performance of region properties, EFD, SimCLR, MAE (trained from scratch and pretrained), O2VAE, and ShapeEmbed on the MNIST and MPEG-7 datasets. We highlight in Table 1 the superior performance of ShapeEmbed over the classical and self-supervised learning baselines, and against its primary competitor. We report additional metrics for the same experiment in Supplementary Section E, which lead to the same conclusions. We hypothesize that the subpar performance of SimCLR is due to the construction of positive pairs, created through image augmentation. To achieve true rotation, scaling, and positional invariance with contrastive learning, one would need to define an alternative way of creating positive pairs of masks that would comprehensively cover all the transformations we normalize for in ShapeEmbed. We stress that these experiments are not meant to push the state-of-the-art in MNIST classification, but instead to evaluate the information content of the shape representation learned by the different methods we consider.

### 4.4 Ablation Studies

**Scaling and Indexation Invariance.** We evaluate the importance of the normalization step and of the various modifications implemented in our VAE to achieve indexation invariance in ShapeEmbed's ability to perform under varying object sizes and contour indexations. To assess the effect of our modified encoder and custom indexation invariant reconstruction loss, we created a modified version of ShapeEmbed in which the circular padding mechanism is replaced by a constant padding of 1 and where the indexation invariant reconstruction loss (2) is substituted with the standard MSE reconstruction loss. To evaluate the effect of normalization, we simply skipped it and retained the original, non-normalized distance matrices. In Table 2, we report F1-scores on a randomly-scaled version of MNIST (sMNIST) and MPEG-7 (sMPEG-7) described in Supplementary Section C. We observe that removing indexation invariance and skipping the normalization step results in a drop in performance on both sMNIST and sMPEG-7, with an even more drastic effect on the latter. These

results illustrate that, when ShapeEmbed does not include scaling and indexation invariance, it captures features in the latent space that are irrelevant to intrinsic shape information and therefore interfere with downstream tasks.

**Rotation and Translation Invariance.** We test the robustness of our model to positional and orientation variations, which are frequently encountered in real-world data. Unlike scaling and indexation invariance, which are explicitly enforced in the model, rotation and translation invariance are inherent to the distance matrix representation we use in ShapeEmbed. Ablating the distance matrix representation thus results in encoding the image mask directly with a vanilla VAE. For the sake of completeness, we also include the performance of O2VAE as a reference, as it partially addresses rotation and translation invariance but still uses masks as input. The results reported in Table 3 illustrate the positive impact of the distance matrix representation. On randomly translated and rotated versions of MNIST and MPEG-7 (rMNIST and rMPEG-7, respectively) as described in Supplementary Section C, ShapeEmbed scores higher than any of the considered alternatives. The gap in performance between ShapeEmbed and the other considered approaches highlights the difficulty of extracting relevant shape features in the absence of explicit translation and rotation invariance in a dataset that exhibits great variability in object orientation and position, and demonstrates the value of the distance matrix representation. We further qualitatively explore the effect of rotation and scaling invariance on the learned representation in Supplementary Section F.

**Further Ablation Studies.** We experimentally explore two more ablation studies in Supplementary Section F: the added value of relying on the VAE latent codes as opposed to using distance matrices directly as shape descriptors, and the effect of the distance matrix regularization terms in the loss.

## 4.5   Application to Biological Imaging

One of the main motivations for this work is its application to biological imaging. Shape, as captured in 2D contours on microscopy images, is one of the most information-rich phenotypic characteristics and provides insights into a range of biological phenomena. Shape analysis in biological images is particularly challenging as objects in these datasets typically appear unaligned, not centered, and may exhibit extensive size variations. Additionally, shape differences in biology often appear as subtle changes, the magnitude and nature of which are typically unknown a priori, making unbiased data exploration invaluable. While experiments usually aim to uncover biological labels (*e.g.*, cell type, cell state), living systems don't come with annotations and researchers only have access to experimental labels (*e.g.*, treated or untreated samples). Using these experimental labels as proxies for the underlying biological labels is inherently problematic due to individual variability: two samples treated identically may respond differently because of natural variations. Self-supervised approaches are especially valuable in approaching this problem as they enable the investigation of biological labels independently of experimental categories.

While scale invariance is traditionally considered to be essential for shape analysis [Dokmanic et al., 2015] and our framework is inherently scale-invariant, scale may be a crucial feature in some applications. To account for this, we additionally consider a variant of ShapeEmbed that preserves scale information by saving the norm of the distance matrix before normalization and concatenating it to the latent code for downstream classification. The decision to consider scale as a relevant feature to be included or as a nuisance transformation to be removed entirely depends on the use case and biological question considered. We assess the value of ShapeEmbed on its own and with size information included (referred to as ShapeEmbed+Sz) on biological imaging datasets at the organism (BBBC010, a well-characterized biological benchmark) and cellular (MEF, a harder, real-life example where the shape component is known to be essential) scales and report performance against the classical and self-supervised learning baselines, as well as against our main competitor, in Table 4. ShapeEmbed consistently outperforms other considered methods, and additional performance can be gained by adding back object size as an extra feature. As objects in the MEF dataset exhibit experimentally-induced size differences between classes in addition to true shape variations, summary statistics, which include size-related metrics (such as the area), perform exceptionally well. This observation highlights the importance of offering a flexible way to handle size information that can adapt to the context. In Supplementary Section G, we report additional metrics for these experiments that lead to the same conclusions. We also qualitatively explore the latent space learned by ShapeEmbed on the BBBC010 and MEF datasets, discuss the robustness of

Table 3: Rotation and translation invariance ablation results (F1-score $\pm \sigma$, higher is better). The input to VAE and O2VAE are binary masks, while ShapeEmbed uses distance matrices as input.

| METHOD | RMNIST | RMPEG |
|---|---|---|
| VAE | $0.38 \pm 0.01$ | $0.04 \pm 0.02$ |
| O2VAE | $0.66 \pm 0.01$ | $0.10 \pm 0.02$ |
| **SHAPEEMBED** | $\mathbf{0.85 \pm 0.01}$ | $\mathbf{0.66 \pm 0.05}$ |

Table 4: Biological imaging datasets results (F1-score $\pm \sigma$, higher is better).

| METHOD | MEF | BBBC010 | HELA KYOTO | MOC |
|---|---|---|---|---|
| REGION PROP. | $0.72 \pm 0.01$ | $0.82 \pm 0.00$ | $0.57 \pm 0.08$ | $0.61 \pm 0.07$ |
| EFD | $0.33 \pm 0.04$ | $0.55 \pm 0.04$ | $0.22 \pm 0.11$ | $0.38 \pm 0.02$ |
| SIMCLR | $0.43 \pm 0.03$ | $0.56 \pm 0.12$ | $0.48 \pm 0.17$ | $0.65 \pm 0.01$ |
| MAE (VIT-B) | $0.54 \pm 0.03$ | $0.60 \pm 0.12$ | $0.47 \pm 0.13$ | $0.57 \pm 0.02$ |
| MAE (VIT-L) | $0.53 \pm 0.02$ | $0.51 \pm 0.07$ | $0.42 \pm 0.09$ | $0.60 \pm 0.07$ |
| MAE (VIT-H) | $0.55 \pm 0.02$ | $0.72 \pm 0.06$ | $0.44 \pm 0.21$ | $0.51 \pm 0.09$ |
| MAE (VIT-B, PRETRAINED) | $0.61 \pm 0.01$ | $0.65 \pm 0.08$ | $0.72 \pm 0.02$ | $0.48 \pm 0.15$ |
| MAE (VIT-L, PRETRAINED) | $0.64 \pm 0.02$ | $0.77 \pm 0.03$ | $0.76 \pm 0.03$ | $0.66 \pm 0.03$ |
| MAE (VIT-H, PRETRAINED) | $0.65 \pm 0.01$ | $0.76 \pm 0.06$ | $0.73 \pm 0.08$ | $0.63 \pm 0.05$ |
| O2VAE | $0.53 \pm 0.02$ | $0.61 \pm 0.08$ | $0.63 \pm 0.08$ | $0.60 \pm 0.06$ |
| SHAPEEMBED | $0.67 \pm 0.01$ | $0.83 \pm 0.00$ | $0.80 \pm 0.04$ | $0.68 \pm 0.01$ |
| **SHAPEEMBED+SZ** | $\mathbf{0.76 \pm 0.01}$ | $\mathbf{0.87 \pm 0.01}$ | $\mathbf{0.85 \pm 0.04}$ | $\mathbf{0.70 \pm 0.05}$ |

our method to the quality of the input segmentation masks, and explore the generative properties of our model.

## 5    Conclusion

We introduced ShapeEmbed, an original self-supervised representation learning framework based on a custom VAE that can, from segmentation masks, extract a latent representation of shape that is agnostic to position, size, orientation, reflection, and contour point indexing. The key ideas behind our method are the use of distance matrices, the implementation of simple but essential modifications to the encoder path of our VAE, and the use of novel loss terms. We experimentally demonstrated the superior performance of ShapeEmbed over existing methods for shape quantification over a range of natural and biological images. We expect ShapeEmbed to be of valuable use for the unbiased exploration of shape variation in image datasets, and expect it to be most impactful in biological imaging where the size, orientation, and position of objects are highly unpredictable and shape differences are subtle. Although ShapeEmbed's requirement for simply-connected contours is arguably restrictive, we have in practice not encountered any case where it cannot be satisfied - either because the objects of interest don't have holes, or because they can be fully defined by a simply-connected midline relying on a ridge detector. Although designed for closed contours, ShapeEmbed can in principle be applied to open curves as well, as we briefly discuss in Supplementary Section H. While our current encoder achieves reflection invariance by processing each matrix as well as as a flipped version, this computational overload could be avoided in the future by using reflection equivariant convolutions [Cohen and Welling, 2016]. Our current implementation uses several regularization terms in the loss to promote the decoded outputs to be valid Euclidean distance matrices, but lacks strict guarantees that the distance matrix constraints are satisfied. An exciting area for future research could be to instead parameterize the predicted matrix in such a way that distance matrix properties are guaranteed to be valid [Dokmanic et al., 2015]. Finally, ShapeEmbed is currently limited to 2D images, but it provides a strong basis for a 3D extension incorporating concepts from the shape signatures literature [Osada et al., 2002] to be explored in future work.

## Acknowledgements

AFR acknowledges funding from the European Molecular Biology Laboratory. CTR acknowledges funding from the European Union's Horizon Europe research and innovation programme under grant agreement 101057970 (AI4Life) and from the European Molecular Biology Laboratory. VU acknowledges funding from the European Molecular Biology Laboratory and from the University of Zurich. The authors thank the members of the Uhlmann group at EMBL-EBI, as well as Anna Kreshuk, Martin Weigert, and Albert Dominguez Mantes for helpful discussions on this work.

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

# ShapeEmbed: a self-supervised learning framework for 2D contour quantification Supplementary Material

Anna Foix Romero[1], Craig Russell[1], Alexander Krull[*2], and Virginie Uhlmann[*1,3]

[1]European Bioinformatics Institute, European Molecular Biology Laboratory, Cambridge, UK,
{afoix, ctr26, uhlmann}@ebi.ac.uk
[2]School of Computer Science, University of Birmingham, Birmingham, UK,
a.f.f.krull@bham.ac.uk
[3]Department of Molecular Life Sciences, University of Zurich, Zurich, CH,
virginie.uhlmann@mls.uzh.ch

[*]Corresponding authors

## A  Distance Matrix Properties

**Translation and Rotation Invariance.** Translation and rotation information is, by design, not captured by distance matrices. Any translation of two points $\mathbf{x}_i, \mathbf{x}_j$ does not affect the Euclidean distance and, consequently, leaves the distance matrix unchanged. This can be formally demonstrated by considering a translation operation that shifts all points in a sequence by a vector $(t_x, t_y)$. The translation operation thus transforms all point $(x, y)$ in the original sequence into $(x', y') = (x + t_x, y + t_y)$. The Euclidean distance between any two points $(x'_i, y'_i)$ and $(x'_j, y'_j)$ in the translated sequence can straightforwardly be shown to be equal to the distance between $(x_i, y_i)$ and $(x_j, y_j)$ in the original sequence as the translation terms cancel out.

Similarly, the distance matrix remains unaffected by rotation. Considering a rotation operation that rotates the points in a sequence by an angle $\theta$ in the counterclockwise direction, the rotation operation transforms all point $(x, y)$ in the original sequence to $(x', y') = (x\cos\theta - y\sin\theta, x\sin\theta + y\cos\theta)$. The Euclidean distance between any two points $(x'_i, y'_i)$ and $(x'_j, y'_j)$ in the rotated sequence can be straightforwardly demonstrated to be equal to the distance between $(x_i, y_i)$ and $(x_j, y_j)$ in the original contour as the trigonometric terms get reduced through the Pythagorean identity.

**Scale Invariance.** Distance matrices are not automatically invariant to scaling. Scaling all points $(x, y)$ in a sequence by a factor $a$ will also scale all distances $d_{i,j}$ by the same factor, thus resulting in a scaled distance matrix $D' = aD$. However, distance matrices can easily be made invariant to scaling upon normalization by the Frobenius matrix norm given by

$$||D||_F = \sqrt{\sum_{i=1}^{N}\sum_{j=1}^{N}(d_{i,j})^2}. \tag{1}$$

The Frobenius norm of a matrix scaled by a factor $a$ is obtained as

$$||aD||_F = a||D||_F, \qquad (2)$$

meaning that scale invariance can be achieved upon normalisation by

$$\bar{D} = \frac{D}{||D||_F}. \qquad (3)$$

## B  Additional Implementation Details

ShapeEmbed is implemented in Python using the PyTorch library ([Paszke et al., 2019], BSD-style license available at `https://github.com/pytorch/pytorch/blob/main/LICENSE`), and is available at `https://github.com/uhlmanngroup/ShapeEmbed`. Experiments were conducted on a machine with an Intel(R) Xeon(R) Gold 6342 CPU @ 2.80GHz and an NVIDIA A100 80GB PCIe GPU. In all experiments, we used the ADAM optimizer [Kingma and Ba, 2015] with a learning rate of $10^{-3}$ for 350 epochs for the datasets we considered. We monitored training using TensorBoard ([Abadi et al., 2015], Apache-2.0 license) and observed that all the models we trained converged appropriately within the considered number of epochs.

All of our experiments were run on a single GPU on our institutional cluster. Runtimes for each experiment are indicated in Section C. The creation of distance matrices from segmentation masks was run on CPUs (Intel(R) Xeon(R) Gold 6252 CPU @ 2.10GHz and Intel(R) Xeon(R) Gold 6336Y CPU @ 2.40GHz) using a single core per run. All performance metrics were computed on CPUs. For the robustness experiments (Section G.3), segmentation with cellpose was run in GPU mode.

## C  Additional Details on the Considered Datasets

We provide example images and corresponding masks for the four bioimaging datasets we consider in Figures 1, 2, 3, and 4. The BBBC010 dataset offers a relatively simple and well-characterized biological benchmark, while the MEF dataset allows us to assess performance on a harder, real-life example where shape information is known to be essential. The MEF dataset is considered to be a faithful representative of a real bioimaging use-case, as testified by its use as a reference in recent works proposing unsupervised frameworks for biological shape analysis [Phillip et al., 2021, Burgess et al., 2024]. The HeLa Kyoto dataset is an interesting example of a small microscopy image dataset comprising a limited number of highly similar objects, where morphological differences are subtle. Finally, the MOC dataset contains cells exhibiting a broad spectrum of morphological variability rangin from smooth to spiky and irregular contours. For this reason, we used a larger distance matrix size ($512 \times 512$) on this dataset to capture details of the cell contours that would otherwise be smoothed out with fewer points.

We provide detailed information on the different datasets used in our experiments in Table 1. In our experiments, we divided each dataset into an $80\%$ / $20\%$ split for training and testing, respectively, relying on stratified sampling [Särndal et al., 2003] to account for class imbalance. Our model architecture dynamically adjusts to arbitrary input distance matrix sizes that are powers of two by determining the number of upsampling steps required. The specific runtimes for each of the considered datasets were as follows: $\sim 49$ hours for MNIST, between 4 and 5 hours for MPEG-7, between 4 and 5 hours for BBBC010, between 7 and 8 hours for MEF, between 22 and 35 minutes for HeLa Kyoto, and between 8 and 9 hours for MOC. Taken together, the number of objects contained in each of these datasets and their size distribution (as indicated in Table 1), along with the reported runtimes, provide a sense of how ShapeEmbed scales with dataset size. The latent space dimension is configurable to balance representational capacity and computational efficiency. By default and in all of our experiments, the latent space is composed of 128 dimensions.

The MNIST and MPEG-7 datasets, in their original form, consist of objects that all have roughly the same size and that have been aligned and centered. For our ablation experiments, we constructed modified versions of the MNIST and MPEG-7 datasets that incorporate size variability through random object scaling (referred to as sMNIST and sMPEG-7 in the main manuscript), as well as positional and rotational variability through random object translation and rotation (referred to as rMNIST and rMPEG-7 in the main manuscript). As a result, objects in these modified datasets neither appear centered nor aligned in the images and exhibit a wide range of different sizes.

Table 1: Summary of the considered datasets and associated experimental settings.

| Dataset | Image Size | Number of Objects | Maximum Outline Size | Minimum Outline Size | Distance Matrix Size | Latent Space Size |
|---|---|---|---|---|---|---|
| MNIST | $28 \times 28$ | 70000 | 177 | 31 | 32 | 128 |
| MPEG-7 | $128 \times 128$ | 1400 | 8049 | 153 | 64 | 128 |
| MEF | $128 \times 128$ | 1407 | 3201 | 63 | 64 | 128 |
| BBBC010 | $64 \times 64$ | 26198 | 409 | 135 | 64 | 128 |
| HeLa Kyoto | $64 \times 64$ | 313 | 248 | 80 | 64 | 128 |
| MOC | $512 \times 512$ | 649 | 3302 | 59 | 512 | 128 |

## D  Baselines Implementation Details

### D.1  Classical Baselines

We run all classical baselines using as input the same outline points from which we build distance matrices for ShapeEmbed.

**Region Properties.** We extract 19 region properties features that pertain to shape using the region-props functionality of the scikit-image library (`https://scikit-image.org/`, MIT license), more specifically the `skimage.measure` module. The 19 measurements we include are all the shape descriptors provided by scikit-image that have no obvious redundancy or localization information. These include the `area`, `convex_area`, `perimeter`, `axis_major_length`, `axis_minor_length`, `extent`, `eccentricity`, `solidity`, `feret_diameter_max`, `hu_moments`, `bbox` and amount to a total to 19 features.

**Elliptical Fourier Descriptors.** We use the `pyefd` library (`https://pyefd.readthedocs.io/en/latest/`, MIT license) to compute Elliptical Fourier Descriptors (EFDs). The Fourier descriptors are extracted using `pyefd.elliptic_fourier_descriptors`, where the number of harmonics in the decomposition is controlled by the order parameter. We set the order to 30, resulting in 120 coefficients per object as each harmonic generates four coefficients (`ana_nan`, `bnb_nbn`, `cnc_ncn`, and `dnd_ndn`). In that way, the EFD feature vector has a comparable number of dimensions to the ShapeEmbed latent space. To ensure invariance to scale, rotation, and translation, the coefficients are normalized using `pyefd.normalize_efd`.

### D.2  Self-Supervised Learning Baselines

We run all self-supervised baselines using as input the same binary masks from which we extract outline points and build distance matrices for ShapeEmbed.

**SimCLR.** We created a SimCLR model with a ResNet18 backbone of 128 output dimensions relying on the original codebase (`https://github.com/sthalles/SimCLR/`, MIT license). We trained for 200 epochs (as we did for ShapeEmbed) using the default configuration and set of transforms to create positive pairs.

**Masked AutoEncoders (MAE).** We benchmarked against the 3 "off-the-shelf" MAE vision transformers (`https://github.com/facebookresearch/mae`, CC-BY-NC 4.0 license) configurations ("base", "large", and "huge", referred to as ViT-b, ViT-l, and ViT-h respectively in our results). We resized the input masks to $224 \times 224$, the input size expected by MAE by default. We used a batch size of 16 (as in the original MAE paper) and 200 epochs (as we did for ShapeEmbed). In addition, we also benchmarked our method against a version of MAE pretrained on ImageNet-1K by initializing the with the PyTorch checkpoints provided at `https://github.com/facebookresearch/mae` (CC-BY-NC 4.0 license).

**O2VAE.** We used the native implementation provided in [Burgess et al., 2024] (`https://github.com/jmhb0/o2vae`, MIT license), running the model with the recommended hyperparameters to ensure consistency and fairness with the published setup. While O2VAE can incorporate both shape and texture information, we here used binary masks as inputs since we specifically focus on shape in our comparison.

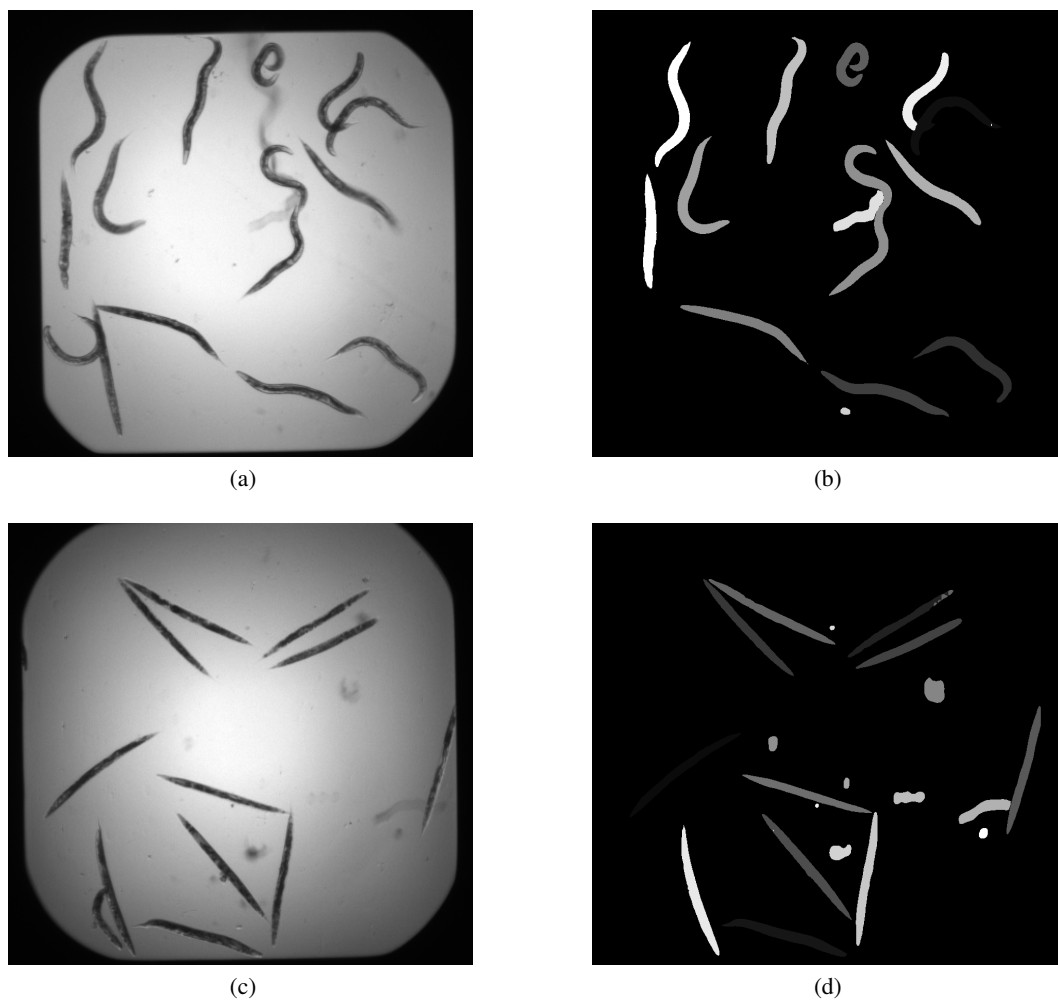

Figure 1: Sample images from the BBBC010 dataset illustrating the (a) live and (c) dead experimental conditions, along with their corresponding instance segmentation masks (b and d).

# E   Additional Benchmarking Results

In addition to the F1-scores reported in the main manuscript in Section 4.3, we report in Tables 2 and 3 the same results with additional significant digits and without rounding, along with the accuracy, precision, recall, and log-loss score as additional metrics to evaluate the performance of our method on the MNIST and MPEG-7 datasets, respectively. The log-loss score quantifies the quality of probabilistic predictions by penalizing incorrect predictions with high confidence [Bishop, 2006]. Higher values of accuracy, precision, and recall, as well as lower log-loss score values, indicate better performance.

The relative performance observed between MAE and ShapeEmbed originates from differences in the representations learned by each method and in the confidence of their predictions. MAE, trained to reconstruct masked portions of input images, capture pixel-level variations in the binary masks. This can lead to embeddings that support highly confident, low-entropy predictions at the classification step and directly drive down the log-loss by assigning high probabilities to the correct class. In contrast, ShapeEmbed encodes structural and geometric properties of the contour through the distance matrix representation, which yields robust and generalizable features that perform well across all reported metrics without optimizing for a specific one.

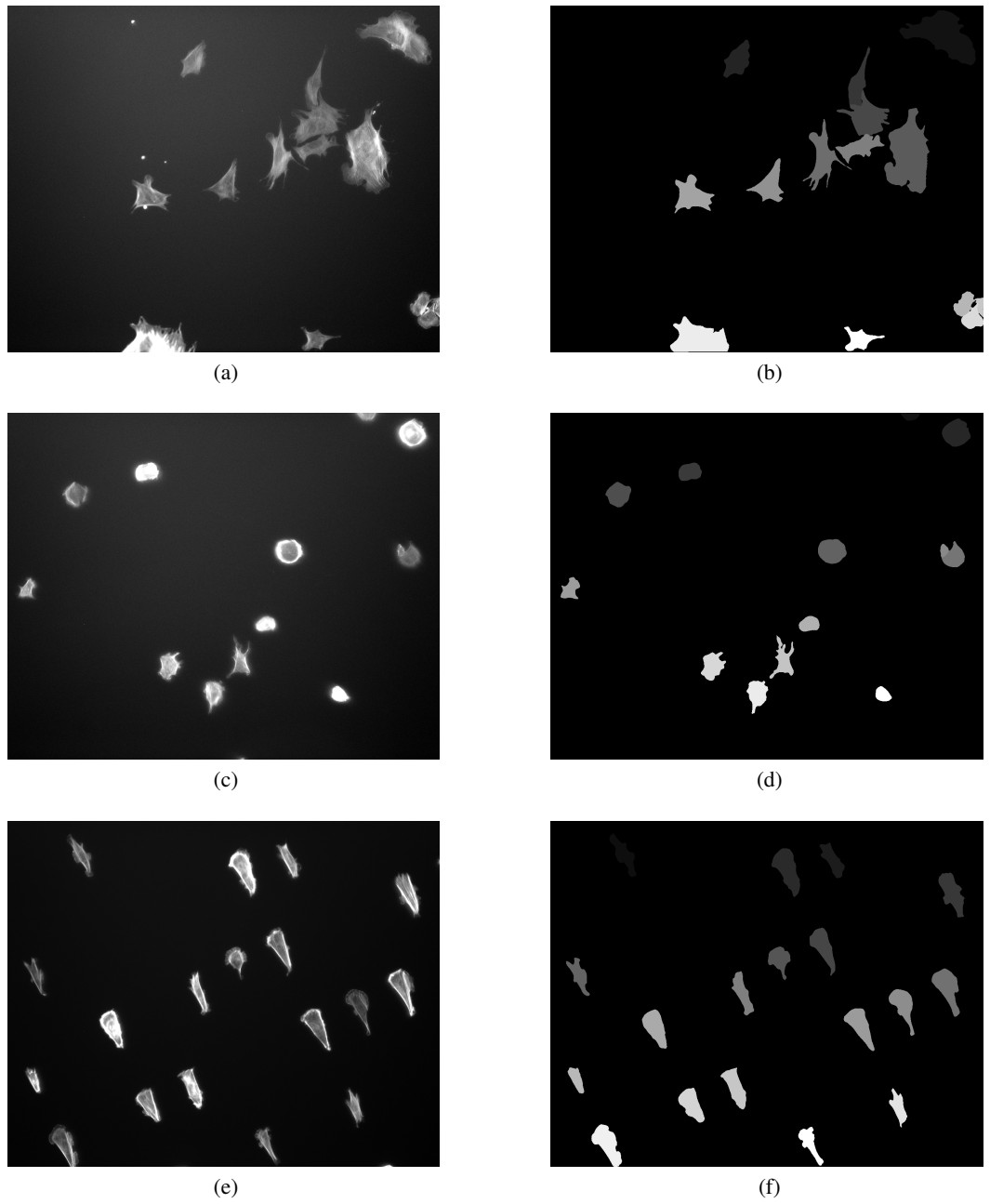

(a)

(b)

(c)

(d)

(e)

(f)

Figure 2: Sample images from the MEF dataset illustrating the (a) control (non-patterned), (c) circle-patterned, and (e) triangle-patterned experimental conditions, along with their corresponding instance segmentation masks (b, d, and f).

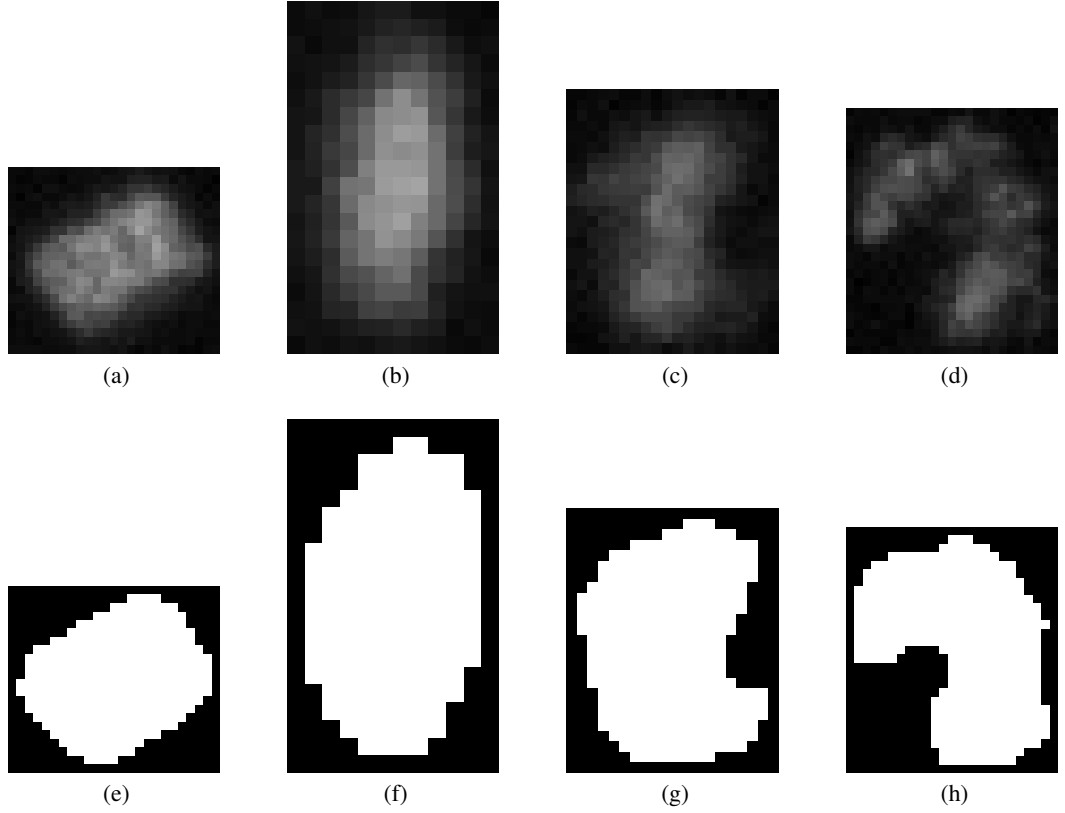

(a)         (b)         (c)         (d)

(e)         (f)         (g)         (h)

Figure 3: Sample images from the HeLa Kyoto dataset illustrating nuclei from the (a) early anaphase, (b) late anaphase, (c) metaphase, (d) prometaphase classes, along with their corresponding instance segmentation masks (e, f, g and h)
.

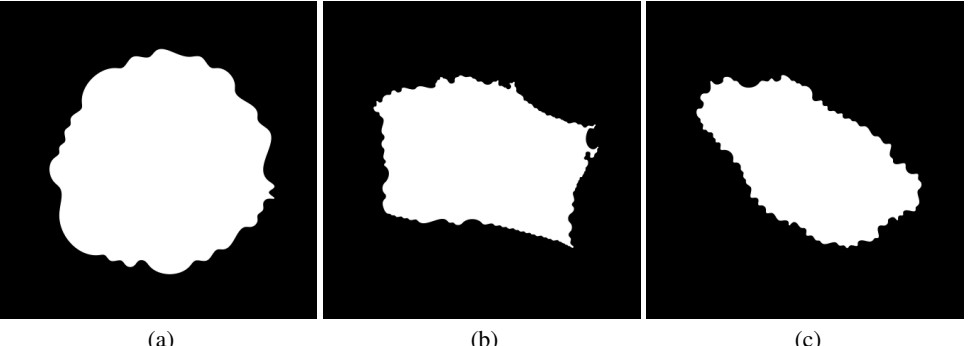

(a)         (b)         (c)

Figure 4: Sample masks generated from the contour coordinates provided in the MOC dataset illustrating the (a) control (non-treated), (c) cytochalasin D-treated (Cytd), and (e) jasplakinolide-treated (Jasp) experimental conditions.

Table 2: Additional performance metrics on on the MNIST dataset.

| METRIC | F1-SCORE | ACCURACY | PRECISION | RECALL | LOG-LOSS |
|---|---|---|---|---|---|
| REGIONS PROP. | $0.809 \pm 0.003$ | $0.801 \pm 0.003$ | $0.808 \pm 0.002$ | $0.807 \pm 0.003$ | $0.675 \pm 0.010$ |
| EFD | $0.623 \pm 0.013$ | $0.621 \pm 0.001$ | $0.631 \pm 0.011$ | $0.630 \pm 0.011$ | $1.005 \pm 0.001$ |
| SIMCLR | $0.593 \pm 0.011$ | $0.598 \pm 0.011$ | $0.594 \pm 0.012$ | $0.598 \pm 0.011$ | $1.188 \pm 0.033$ |
| MAE (VIT-B) | $0.953 \pm 0.026$ | $0.953 \pm 0.003$ | $0.953 \pm 0.003$ | $0.952 \pm 0.004$ | $\mathbf{0.062 \pm 0.020}$ |
| MAE (VIT-L) | $0.840 \pm 0.009$ | $0.841 \pm 0.009$ | $0.841 \pm 0.009$ | $0.840 \pm 0.009$ | $0.520 \pm 0.021$ |
| MAE (VIT-H) | $0.848 \pm 0.007$ | $0.921 \pm 0.004$ | $0.919 \pm 0.003$ | $0.921 \pm 0.004$ | $0.369 \pm 0.018$ |
| MAE (VIT-B, PRETRAINED) | $0.921 \pm 0.009$ | $0.921 \pm 0.007$ | $0.930 \pm 0.010$ | $0.912 \pm 0.006$ | $0.211 \pm 0.009$ |
| MAE (VIT-L, PRETRAINED) | $0.934 \pm 0.013$ | $0.935 \pm 0.013$ | $0.940 \pm 0.003$ | $0.929 \pm 0.007$ | $0.199 \pm 0.007$ |
| MAE (VIT-H, PRETRAINED) | $0.942 \pm 0.012$ | $0.941 \pm 0.002$ | $0.946 \pm 0.004$ | $0.938 \pm 0.001$ | $0.204 \pm 0.017$ |
| O2VAE | $0.859 \pm 0.007$ | $0.859 \pm 0.008$ | $0.860 \pm 0.008$ | $0.859 \pm 0.008$ | $0.717 \pm 0.043$ |
| **SHAPEEMBED** | $\mathbf{0.963 \pm 0.007}$ | $\mathbf{0.961 \pm 0.005}$ | $\mathbf{0.964 \pm 0.009}$ | $\mathbf{0.964 \pm 0.008}$ | $0.187 \pm 0.020$ |

Table 3: Additional performance metrics on the MPEG-7 dataset.

| METRIC | F1-SCORE | ACCURACY | PRECISION | RECALL | LOG-LOSS |
|---|---|---|---|---|---|
| REGION PROP. | $0.701 \pm 0.014$ | $0.724 \pm 0.002$ | $0.714 \pm 0.003$ | $0.688 \pm 0.027$ | $1.257 \pm 0.113$ |
| EFD | $0.079 \pm 0.008$ | $0.077 \pm 0.003$ | $0.076 \pm 0.006$ | $0.076 \pm 0.008$ | $2.909 \pm 0.015$ |
| SIMCLR | $0.128 \pm 0.020$ | $0.141 \pm 0.016$ | $0.145 \pm 0.022$ | $0.141 \pm 0.016$ | $22.502 \pm 0.522$ |
| MAE (VIT-B) | $0.646 \pm 0.001$ | $0.675 \pm 0.024$ | $0.660 \pm 0.016$ | $0.675 \pm 0.024$ | $1.471 \pm 0.071$ |
| MAE (VIT-L) | $0.627 \pm 0.040$ | $0.654 \pm 0.037$ | $0.637 \pm 0.037$ | $0.654 \pm 0.037$ | $1.465 \pm 0.112$ |
| MAE (VIT-H) | $0.600 \pm 0.010$ | $0.633 \pm 0.166$ | $0.615 \pm 0.001$ | $0.601 \pm 0.045$ | $1.767 \pm 0.079$ |
| MAE (VIT-B, PRETRAINED) | $0.422 \pm 0.011$ | $0.436 \pm 0.010$ | $0.400 \pm 0.115$ | $0.447 \, pm 0.095$ | $1.887 \pm 0.084$ |
| MAE (VIT-L, PRETRAINED) | $0.512 \pm 0.019$ | $0.524 \pm 0.008$ | $0.518 \pm 0.030$ | $0.516 \pm 0.010$ | $1.723 \pm 0.082$ |
| MAE (VIT-H, PRETRAINED) | $0.571 \pm 0.021$ | $0.574 \pm 0.011$ | $0.580 \pm 0.020$ | $0.565 \pm 0.021$ | $1.812 \pm 0.079$ |
| O2VAE | $0.128 \pm 0.020$ | $0.566 \pm 0.008$ | $0.538 \pm 0.010$ | $0.566 \pm 0.008$ | $2.891 \pm 0.070$ |
| **SHAPEEMBED** | $\mathbf{0.751 \pm 0.024}$ | $\mathbf{0.763 \pm 0.037}$ | $\mathbf{0.716 \pm 0.002}$ | $\mathbf{0.763 \pm 0.036}$ | $\mathbf{1.158 \pm 0.206}$ |

# F    Additional Ablation Results

## F.1    Effect of Rotation and Scaling Invariance on the Learned Representation

To complement Section 4.4 of the main manuscript and further qualitatively explore the effect of rotation and scaling invariance on the learned representation, we generated 2D projections of the latent space learned by the vanilla VAE and by ShapeEmbed relying on the t-SNE [van der Maaten and Hinton, 2008] dimensionality reduction technique. We display the t-SNE projections of the rMNIST latent space in Figure 5, where individual data points are colored according to the class label of their original input image. We observe that the latent representation learned by the vanilla VAE is randomly structured and does not allow resolving individual classes. The latent representation learned by ShapeEmbed, however, aggregates data points with similar class labels together, as one would expect the vanilla VAE to behave on the standard MNIST dataset composed of pre-aligned and centered objects. The t-SNE algorithm is used with a random seed of $42$ and a perplexity of $5$, which are commonly used default parameters.

## F.2    Added Value of the Representation Learning Model

Our motivation for encoding distance matrices with a VAE is that the ShapeEmbed model will distill these input structures into a compact and highly informative latent representation. To verify that the VAE encoding step genuinely results in better shape descriptors, we compare the classification performance obtained with ShapeEmbed latent codes against that obtained when relying on raw distance matrices directly in our classifier. The results presented in Tables 4, 5, 6, 7, and 8 consistently demonstrate that the latent representation learned by ShapeEmbed allows for better shape discrimination over the MNIST, MPEG-7, BBBC010, and MEF datasets.

## F.3    Distance Matrix Regularization and Outline Reconstruction

The distance matrix regularization loss terms introduced in Section 3.3 of the main manuscript play an important role in ensuring that the latent space learned by ShapeEmbed captures the structure of the distance matrices while maintaining reconstruction fidelity. To assess the impact of these custom regularization loss terms on downstream classification performance, we perform ablation experiments in which we remove each of the regularization terms one by one. Results on the MNIST, MPEG-7,

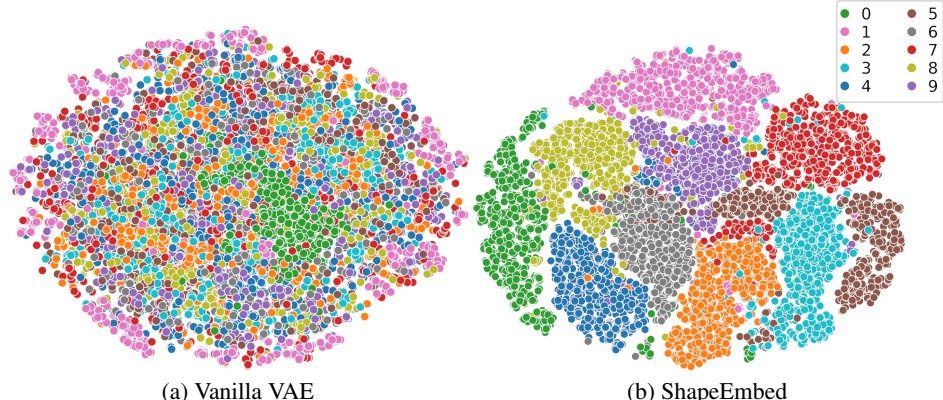

(a) Vanilla VAE                                          (b) ShapeEmbed

Figure 5: Projection (t-SNE) of the rMNIST latent space. (a) The latent representation of rMNIST learned by a vanilla VAE does not exhibit any noticeable structure and class separation. In contrast, (b) the latent representation of rMNIST learned by ShapeEmbed, which ignores orientation and position, recovers clusters of data points that match their underlying class.

Table 4: Effect of the VAE encoding on classification performance. Mean and standard deviation of the F1-score over 5-fold cross-validation. Higher values indicate better performance.

| DATASET | DISTANCE MATRICES | SHAPEEMBED |
|---------|-------------------|------------|
| MNIST   | $0.912 \pm 0.011$ | $\mathbf{0.963 \pm 0.007}$ |
| MPEG-7  | $0.377 \pm 0.038$ | $\mathbf{0.751 \pm 0.024}$ |
| BBBC010 | $0.737 \pm 0.025$ | $\mathbf{0.866 \pm 0.008}$ |
| MEF     | $0.299 \pm 0.006$ | $\mathbf{0.746 \pm 0.006}$ |

Table 5: Effect of the VAE encoding on classification performance. Mean and standard deviation of the accuracy over 5-fold cross-validation. Higher values indicate better performance.

| DATASET | DISTANCE MATRICES | SHAPEEMBED |
|---------|-------------------|------------|
| MNIST   | $0.915 \pm 0.009$ | $\mathbf{0.961 \pm 0.005}$ |
| MPEG-7  | $0.378 \pm 0.029$ | $\mathbf{0.763 \pm 0.037}$ |
| BBBC010 | $0.737 \pm 0.025$ | $\mathbf{0.831 \pm 0.003}$ |
| MEF     | $0.343 \pm 0.007$ | $\mathbf{0.670 \pm 0.009}$ |

Table 6: Effect of the VAE encoding on classification performance. Mean and standard deviation of the precision over 5-fold cross-validation. Higher values indicate better performance.

| DATASET | DISTANCE MATRICES | SHAPEEMBED |
|---------|-------------------|------------|
| MNIST   | $0.911 \pm 0.008$ | $\mathbf{0.964 \pm 0.009}$ |
| MPEG-7  | $0.378 \pm 0.022$ | $\mathbf{0.716 \pm 0.002}$ |
| BBBC010 | $0.737 \pm 0.025$ | $\mathbf{0.811 \pm 0.011}$ |
| MEF     | $0.452 \pm 0.024$ | $\mathbf{0.674 \pm 0.004}$ |

BBBC010, and MEF datasets shown in Table 9 illustrate that performance is not heavily affected by the absence of distance matrix regularization terms, which is expected as the classification task relies on the latent representations and does not exploit the outline reconstruction.

Reconstructions of a good enough quality may, however, be of strong interest in tasks that focus on generative modeling or inverse mapping from the latent space. This motivates the inclusion of the regularization terms, as demonstrated in Figure 6, where we visually illustrate the impact of removing the distance matrix regularization loss terms on the quality of the reconstructed outlines in the MNIST and MPEG-7 datasets. Even though downstream classification performance is only negligibly affected

Table 7: Effect of the VAE encoding on classification performance. Mean and standard deviation of the recall over 5-fold cross-validation. Higher values indicate better performance.

| DATASET | DISTANCE MATRICES | SHAPEEMBED |
|---------|-------------------|------------|
| MNIST   | $0.911 \pm 0.007$ | $\mathbf{0.964 \pm 0.008}$ |
| MPEG-7  | $0.379 \pm 0.020$ | $\mathbf{0.763 \pm 0.036}$ |
| BBBC010 | $0.734 \pm 0.026$ | $\mathbf{0.812 \pm 0.009}$ |
| MEF     | $0.343 \pm 0.007$ | $\mathbf{0.670 \pm 0.009}$ |

Table 8: Effect of the VAE encoding on classification performance. Mean and standard deviation of the log-loss over 5-fold cross-validation. Lower values indicate better performance.

| DATASET | DISTANCE MATRICES | SHAPEEMBED |
|---------|-------------------|------------|
| MNIST   | $0.371 \pm 0.018$ | $\mathbf{0.187 \pm 0.020}$ |
| MPEG-7  | $1.201 \pm 0.013$ | $\mathbf{1.158 \pm 0.206}$ |
| BBBC010 | $2.889 \pm 0.686$ | $\mathbf{0.640 \pm 0.016}$ |
| MEF     | $1.202 \pm 0.066$ | $\mathbf{0.801 \pm 0.019}$ |

Table 9: Effect of the distance matrix regularization loss terms on classification performance. Mean and standard deviation of the F1-score over 5-fold cross-validation. Higher values indicate better performance.

| METHOD | MNIST | MPEG-7 | BBBC010 | MEF |
|--------|-------|--------|---------|-----|
| NO SYMMETRY | $0.941 \pm 0.006$ | $0.536 \pm 0.057$ | $0.769 \pm 0.015$ | $0.681 \pm 0.034$ |
| NO DIAGONAL | $0.952 \pm 0.005$ | $0.604 \pm 0.067$ | $0.801 \pm 0.021$ | $0.733 \pm 0.044$ |
| NO NON-NEGATIVITY | $0.923 \pm 0.007$ | $0.581 \pm 0.037$ | $0.789 \pm 0.011$ | $0.701 \pm 0.052$ |
| **SHAPEEMBED** | $\mathbf{0.963 \pm 0.007}$ | $\mathbf{0.751 \pm 0.024}$ | $\mathbf{0.831 \pm 0.003}$ | $\mathbf{0.760 \pm 0.008}$ |

by the ablation of these regularization terms, the reconstructions without regularization are noticeably degraded.

As ShapeEmbed relies on a VAE model, it allows for reconstructing outlines from their latent codes, but can in addition also generate outlines from vectors that have been sampled in the latent space. We illustrate in Figure 7 examples of randomly-picked samples from different classes of the MPEG-7 dataset along with the mean outline of each of these classes. To generate the mean outlines, we compute the average latent representation of each class and decode it back into an outline in image space. Specifically, for each unique label in the dataset, we extract the latent vectors of all objects having that label and compute their mean. This averaged latent vector is then sent through the model's decoder to reconstruct the mean outline. We observe that the mean outlines correctly match the intuition of the average shape of each class, despite the various sizes and orientations of individual objects within the classes.

Furthermore, we can also generate outlines by randomly sampling the latent space. To do so, we retrain our model and set the hyperparameter $\beta$ to $10^{-5}$, thus enforcing a smoother and more structured latent space that aligns with a normal prior. We then generate random latent vectors by sampling from a normal distribution with a mean of zero and a standard deviation of one. The sampled vectors are finally sent through the decoder to reconstruct their corresponding outlines. We illustrate in Figure 8 examples of outlines generated in this manner from the latent space learned by ShapeEmbed on the MNIST dataset. We observe that the generated outlines form plausible shapes of the different digits present in MNIST.

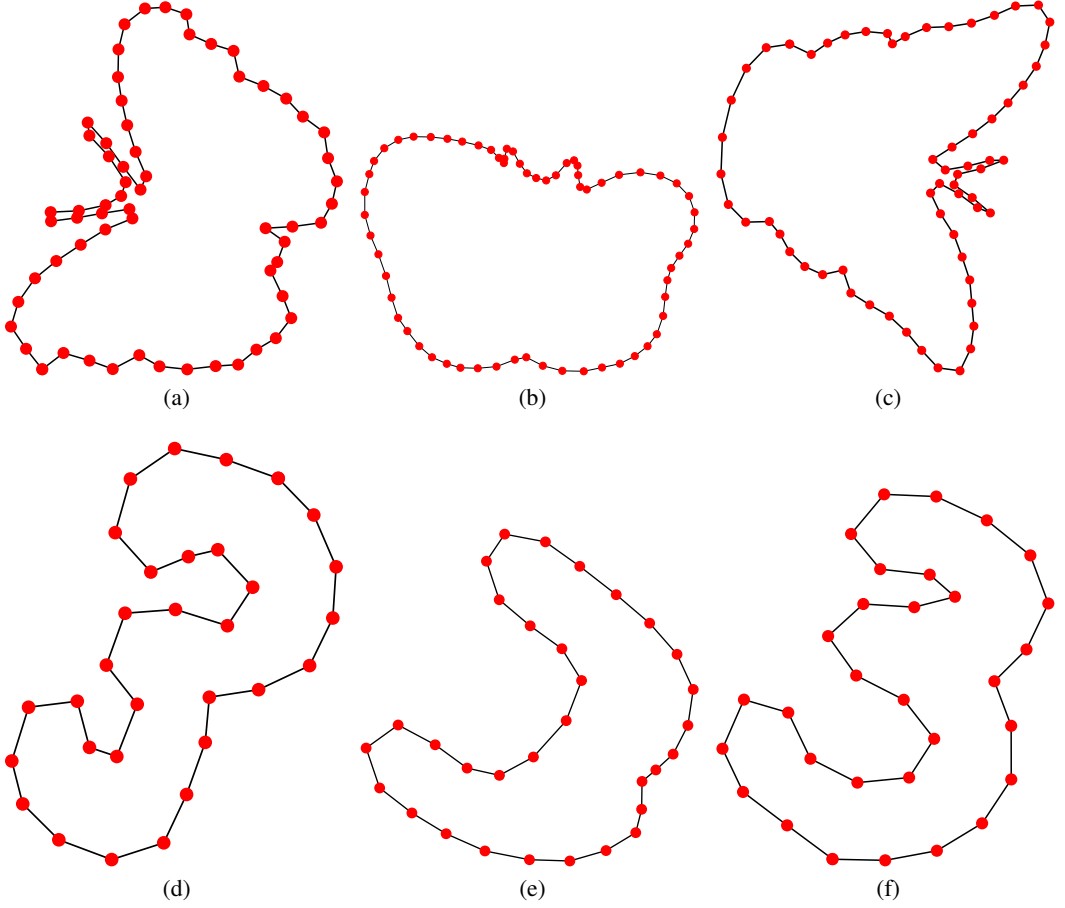

Figure 6: Effect of distance matrix regularization on outline reconstruction. For a randomly-picked sample of the MPEG-7 and MNIST datasets, we show the original outline (a and d), the reconstructed outline without any of the distance matrix regularization terms (b and e), and the reconstructed outline with all distance matrix regularization terms (c and f).

## G    Additional Biological Imaging Results

### G.1    Additional Performance Metrics

In addition to the F1-scores reported in the main manuscript in Section 4.3, we report in Tables 10, 11, 12, and 13 the same results with additional significant digits and without rounding, along with the accuracy, precision, recall, and log-loss score as additional metrics to evaluate the performance of our method on the BBBC010, MEF, HeLa Kyoto, and MOC datasets, respectively. As space allows, we here include additional significant digits in the results we report. Higher values of accuracy, precision, and recall, as well as lower log-loss score values, indicate better performance. As in the main manuscript, we include results obtained with ShapeEmbed directly, as well as those obtained by adding size (*i.e.*, the distance matrix norm) back as an extra feature. This flexibility is motivated by the fact that, in biological imaging, size invariance may either be a crucial or entirely irrelevant feature depending on the context: it is necessary when size differences arise from imaging conditions (such as varying magnifications) but undesired when size differences are biologically meaningful (such as varying growth rate).

It is important to note that BBBC010 is not a *pure* shape dataset: it originally also includes intensity and texture information. We here do not use BBBC010 to demonstrate that we provide the best results possible when attempting to classify this dataset, but instead to show that we can learn, without any supervision, a good representation of the biological shapes it contains in a way that allows for

Randomly-picked samples          Mean outline

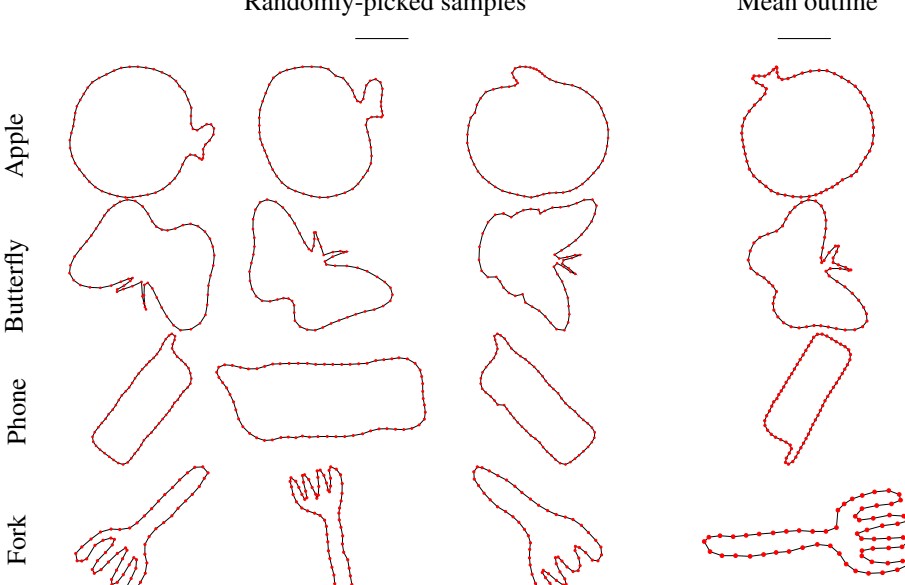

Figure 7: Mean outline generation in the MPEG-7 dataset. We illustrate 3 randomly-picked samples in 4 of the classes of the MPEG-7 dataset (first three columns), along with the mean outline obtained by decoding the vector corresponding to the average of each of the classes in the ShapeEmbed latent space (last column).

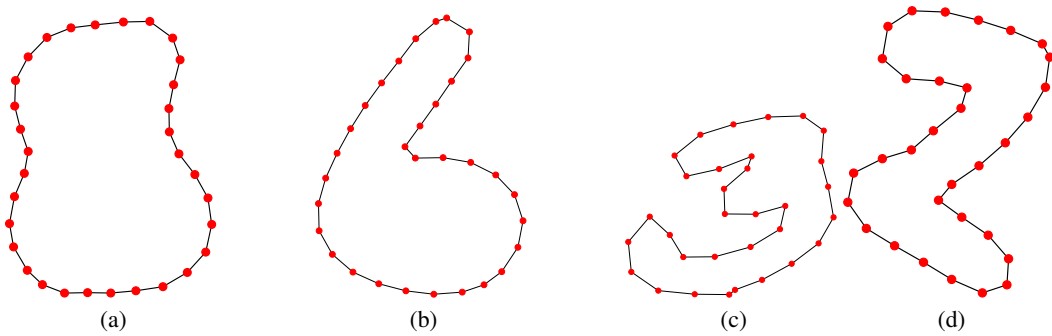

(a)                    (b)                    (c)                    (d)

Figure 8: Novel outline generation in the MNIST dataset. We illustrate 4 outlines obtained by randomly sampling vectors from a normal distribution in the ShapeEmbed latent space that are subsequently reconstructed through the decoder path.

the unbiased exploration of the data. This motivates the comparison against a shape-only baseline (Region Prop.) in Table 10 as opposed to the method proposed in Wählby et al. [2012], which reports a higher classification accuracy on this dataset by leveraging intensity and texture features.

We observe that region properties obtain a lower log-loss on the BBBC010 dataset despite underperforming according to all other metrics relative to the version of ShapeEmbed that includes object size. The log-loss assigns higher penalties for incorrect predictions that have a high certainty. From this, we hypothesize that ShapeEmbed obtains a higher F1-score because the latent codes it produces result in more confident and more correct predictions, but also overconfident errors. As a consequence, ShapeEmbed obtains a slightly higher log-loss when compared to region properties.

Table 10: Additional performance metrics on the BBBC010 dataset.

| METRIC | F1-SCORE | ACCURACY | PRECISION | RECALL | LOG-LOSS |
|---|---|---|---|---|---|
| REGION PROP. | $0.821 \pm 0.002$ | $0.824 \pm 0.022$ | $0.826 \pm 0.021$ | $0.824 \pm 0.022$ | $\mathbf{0.412 \pm 0.027}$ |
| EFD | $0.547 \pm 0.038$ | $0.579 \pm 0.031$ | $0.574 \pm 0.039$ | $0.579 \pm 0.031$ | $0.682 \pm 0.009$ |
| SIMCLR | $0.562 \pm 0.117$ | $0.567 \pm 0.115$ | $0.569 \pm 0.119$ | $0.567 \pm 0.115$ | $0.762 \pm 0.125$ |
| MAE (VIT-B) | $0.597 \pm 0.119$ | $0.628 \pm 0.105$ | $0.633 \pm 0.107$ | $0.628 \pm 0.105$ | $0.716 \pm 0.156$ |
| MAE (VIT-L) | $0.514 \pm 0.072$ | $0.720 \pm 0.580$ | $0.723 \pm 0.058$ | $0.721 \pm 0.060$ | $0.632 \pm 0.082$ |
| MAE (VIT-H) | $0.718 \pm 0.059$ | $0.657 \pm 0.081$ | $0.671 \pm 0.090$ | $0.657 \pm 0.081$ | $0.649 \pm 0.083$ |
| MAE (VIT-B, PRETRAINED) | $0.649 \pm 0.081$ | $0.652 \pm 0.017$ | $0.654 \pm 0.005$ | $0.657 \pm 0.009$ | $0.711 \pm 0.075$ |
| MAE (VIT-L, PRETRAINED) | $0.771 \pm 0.031$ | $0.773 \pm 0.002$ | $0.790 \pm 0.006$ | $0.753 \pm 0.014$ | $0.650 \pm 0.045$ |
| MAE (VIT-H, PRETRAINED) | $0.762 \pm 0.059$ | $0.768 \pm 0.008$ | $0.765 \pm 0.022$ | $0.764 \pm 0.007$ | $0.699 \pm 0.017$ |
| O2VAE | $0.605 \pm 0.080$ | $0.602 \pm 0.071$ | $0.630 \pm 0.084$ | $0.624 \pm 0.712$ | $0.661 \pm 0.020$ |
| SHAPEEMBED | $0.831 \pm 0.003$ | $0.831 \pm 0.003$ | $0.811 \pm 0.011$ | $0.812 \pm 0.009$ | $0.640 \pm 0.016$ |
| **SHAPEEMBED + SIZE** | $\mathbf{0.871 \pm 0.008}$ | $\mathbf{0.872 \pm 0.015}$ | $\mathbf{0.866 \pm 0.009}$ | $\mathbf{0.866 \pm 0.009}$ | $0.509 \pm 0.136$ |

Table 11: Additional performance metrics on the MEF dataset.

| METRIC | F1-SCORE | ACCURACY | PRECISION | RECALL | LOG-LOSS |
|---|---|---|---|---|---|
| REGION PROP. | $0.722 \pm 0.006$ | $0.722 \pm 0.006$ | $0.732 \pm 0.007$ | $0.723 \pm 0.008$ | $0.649 \pm 0.011$ |
| EFD | $0.326 \pm 0.041$ | $0.403 \pm 0.002$ | $0.324 \pm 0.005$ | $0.353 \pm 0.008$ | $1.051 \pm 0.001$ |
| SIMCLR | $0.434 \pm 0.031$ | $0.444 \pm 0.029$ | $0.451 \pm 0.032$ | $0.444 \pm 0.029$ | $1.019 \pm 0.020$ |
| MAE (VIT-B) | $0.537 \pm 0.030$ | $0.537 \pm 0.031$ | $0.539 \pm 0.030$ | $0.546 \pm 0.029$ | $0.895 \pm 0.024$ |
| MAE (VIT-L) | $0.532 \pm 0.019$ | $0.535 \pm 0.019$ | $0.534 \pm 0.020$ | $0.535 \pm 0.018$ | $0.885 \pm 0.028$ |
| MAE (VIT-H) | $0.549 \pm 0.023$ | $0.549 \pm 0.023$ | $0.552 \pm 0.024$ | $0.549 \pm 0.023$ | $0.830 \pm 0.034$ |
| MAE (VIT-B, PRETRAINED) | $0.611 \pm 0.013$ | $0.619 \pm 0.020$ | $0.609 \pm 0.005$ | $0.626 \pm 0.001$ | $0.702 \pm 0.008$ |
| MAE (VIT-L, PRETRAINED) | $0.643 \pm 0.024$ | $0.651 \pm 0.013$ | $0.631 \pm 0.003$ | $0.656 \pm 0.015$ | $0.688 \pm 0.033$ |
| MAE (VIT-H, PRETRAINED) | $0.651 \pm 0.009$ | $0.672 \pm 0.011$ | $0.656 \pm 0.002$ | $0.653 \pm 0.006$ | $0.699 \pm 0.033$ |
| O2VAE | $0.527 \pm 0.023$ | $0.527 \pm 0.024$ | $0.528 \pm 0.023$ | $0.527 \pm 0.024$ | $1.191 \pm 0.108$ |
| SHAPEEMBED | $0.670 \pm 0.009$ | $0.670 \pm 0.009$ | $0.674 \pm 0.004$ | $0.670 \pm 0.009$ | $0.801 \pm 0.019$ |
| **SHAPEEMBED + SIZE** | $\mathbf{0.760 \pm 0.008}$ | $\mathbf{0.755 \pm 0.006}$ | $\mathbf{0.751 \pm 0.006}$ | $\mathbf{0.753 \pm 0.005}$ | $\mathbf{0.640 \pm 0.016}$ |

Table 12: Additional performance metrics on the HeLa Kyoto dataset.

| METRIC | F1-SCORE | ACCURACY | PRECISION | RECALL | LOG-LOSS |
|---|---|---|---|---|---|
| REGION PROP. | $0.569 \pm 0.076$ | $0.578 \pm 0.066$ | $0.554 \pm 0.050$ | $0.594 \pm 0.003$ | $0.782 \pm 0.095$ |
| EFD | $0.221 \pm 0.110$ | $0.255 \pm 0.125$ | $0.223 \pm 0.103$ | $0.221 \pm 0.109$ | $2.003 \pm 0.036$ |
| SIMCLR | $0.482 \pm 0.171$ | $0.509 \pm 0.013$ | $0.470 \pm 0.015$ | $0.494 \pm 0.009$ | $1.069 \pm 0.044$ |
| MAE (VIT-B) | $0.472 \pm 0.002$ | $0.488 \pm 0.001$ | $0.471 \pm 0.002$ | $0.477 \pm 0.003$ | $1.002 \pm 0.101$ |
| MAE (VIT-L) | $0.422 \pm 0.009$ | $0.435 \pm 0.008$ | $0.422 \pm 0.008$ | $0.419 \pm 0.009$ | $1.053 \pm 0.099$ |
| MAE (VIT-H) | $0.444 \pm 0.001$ | $0.449 \pm 0.007$ | $0.446 \pm 0.001$ | $0.444 \pm 0.001$ | $1.042 \pm 0.008$ |
| MAE (VIT-B, PRETRAINED) | $0.721 \pm 0.002$ | $0.728 \pm 0.005$ | $0.721 \pm 0.002$ | $0.728 \pm 0.005$ | $0.659 \pm 0.011$ |
| MAE (VIT-L, PRETRAINED) | $0.761 \pm 0.003$ | $0.770 \pm 0.011$ | $0.761 \pm 0.003$ | $0.765 \pm 0.001$ | $0.699 \pm 0.017$ |
| MAE (VIT-H, PRETRAINED) | $0.732 \pm 0.001$ | $0.741 \pm 0.017$ | $0.732 \pm 0.001$ | $0.733 \pm 0.004$ | $0.653 \pm 0.005$ |
| O2VAE | $0.630 \pm 0.082$ | $0.639 \pm 0.025$ | $0.630 \pm 0.082$ | $0.635 \pm 0.062$ | $0.699 \pm 0.018$ |
| SHAPEEMBED | $0.801 \pm 0.041$ | $0.811 \pm 0.015$ | $0.801 \pm 0.041$ | $0.810 \pm 0.361$ | $0.639 \pm 0.026$ |
| **SHAPEEMBED + SIZE** | $\mathbf{0.854 \pm 0.044}$ | $\mathbf{0.854 \pm 0.010}$ | $\mathbf{0.854 \pm 0.044}$ | $\mathbf{0.853 \pm 0.024}$ | $\mathbf{0.601 \pm 0.013}$ |

Table 13: Additional performance metrics on the MOC dataset.

| METRIC | F1-SCORE | ACCURACY | PRECISION | RECALL | LOG-LOSS |
|---|---|---|---|---|---|
| REGION PROP. | $0.604 \pm 0.074$ | $0.606 \pm 0.011$ | $0.599 \pm 0.034$ | $0.609 \pm 0.074$ | $0.816 \pm 0.007$ |
| EFD | $0.382 \pm 0.021$ | $0.386 \pm 0.106$ | $0.377 \pm 0.021$ | $0.387 \pm 0.008$ | $1.093 \pm 0.014$ |
| SIMCLR | $0.650 \pm 0.014$ | $0.653 \pm 0.022$ | $0.647 \pm 0.014$ | $0.653 \pm 0.011$ | $0.721 \pm 0.065$ |
| MAE (VIT-B) | $0.571 \pm 0.003$ | $0.581 \pm 0.007$ | $0.566 \pm 0.003$ | $0.576 \pm 0.002$ | $0.788 \pm 0.100$ |
| MAE (VIT-L) | $0.598 \pm 0.001$ | $0.601 \pm 0.002$ | $0.596 \pm 0.003$ | $0.600 \pm 0.001$ | $0.746 \pm 0.054$ |
| MAE (VIT-H) | $0.513 \pm 0.001$ | $0.531 \pm 0.006$ | $0.504 \pm 0.006$ | $0.522 \pm 0.001$ | $0.965 \pm 0.006$ |
| MAE (VIT-B, PRETRAINED) | $0.481 \pm 0.007$ | $0.499 \pm 0.004$ | $0.472 \pm 0.007$ | $0.472 \pm 0.007$ | $1.003 \pm 0.034$ |
| MAE (VIT-L, PRETRAINED) | $0.664 \pm 0.002$ | $0.672 \pm 0.002$ | $0.660 \pm 0.005$ | $0.668 \pm 0.002$ | $0.622 \pm 0.022$ |
| MAE (VIT-H, PRETRAINED) | $0.631 \pm 0.051$ | $0.638 \pm 0.003$ | $0.628 \pm 0.051$ | $0.634 \pm 0.021$ | $0.686 \pm 0.076$ |
| O2VAE | $0.595 \pm 0.026$ | $0.599 \pm 0.003$ | $0.593 \pm 0.026$ | $0.590 \pm 0.016$ | $0.816 \pm 0.066$ |
| SHAPEEMBED | $0.682 \pm 0.012$ | $0.699 \pm 0.004$ | $0.673 \pm 0.012$ | $0.683 \pm 0.009$ | $0.649 \pm 0.008$ |
| **SHAPEEMBED + SIZE** | $0.703 \pm 0.051$ | $0.711 \pm 0.002$ | $0.699 \pm 0.051$ | $0.707 \pm 0.011$ | $0.631 \pm 0.001$ |

## G.2  Qualitative Exploration of the Latent Space

Further to quantitative classification results, we also qualitatively explore the latent space learned by ShapeEmbed on the BBBC010 and MEF datasets through the 2D t-SNE projection displayed in Figures 9 and 10 and obtained with the same parameters as Figure 5.

In Figure 9, individual data points are colored according to the class label of their original input image in the BBBC010 dataset, which is either dead or alive. In BBBC010, labels have been derived from experimental conditions (whether the sample has been treated by a lethal substance or not). When dead, *C. elegans* nematodes straighten to look like a rod, while they swim sinusoidally and curve when alive. ShapeEmbed is not only able to identify these distinct shape populations without supervision, but upon inspection of the structure of the latent space, it also reveals instances where experimental labels do not align with biological reality. Several of the "misclassified" data points indeed correspond to mislabeled nematodes that are either alive despite having been treated or dead despite being untreated. This interesting finding illustrates that classification performance alone is not a good indicator of the ability to distinguish between biological states in the BBBC010 dataset, as some nematodes labeled live appear to be dead and vice versa.

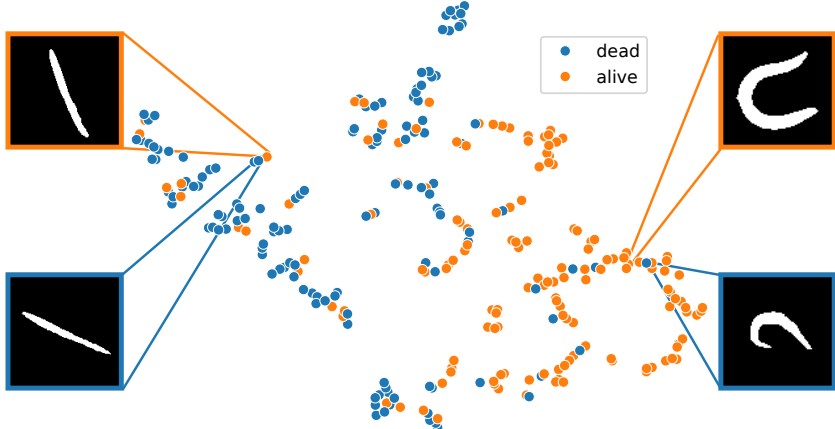

Figure 9: Projection (t-SNE) of the BBBC010 latent space learned by ShapeEmbed. Data points are grouped according to their corresponding classes, namely dead (straight rods) and live (curved worms) nematodes. A closer inspection of data points that seem to be misplaced reveals that their associated class label does not reflect their actual shape.

In Figure 10, individual data points are colored according to the class label of their original input image in the MEF dataset, which is either "circle", "triangle", or "control". The MEF dataset contains images of cells that were cultured on fibronectin micropattern surfaces to enforce cell shape constraints, as described in Hale et al. [2011]. This real use case illustrates the challenge of untangling individual biological variability from experimental variability and has been the dataset of choice in recent papers proposing unsupervised frameworks for biological shape analysis [Phillip et al., 2021, Burgess et al., 2024]. ShapeEmbed is capable of separating the cells grown on circle- and triangle-patterned surfaces, but in addition reveals that cells in the control class, which are not cultured on a patterned surface, adopt diverse shapes that overlap with the two other classes. This is confirmed by a closer inspection of the masks: cells cultured on an unpatterned surface may naturally exhibit the whole range of shapes observed in cells cultured on patterned surfaces. This observation illustrates that relying on a structured latent space to investigate the distributions of shape phenotypes in biology often provides richer information than relying on the output of a classifier, and is therefore more likely to provide insights into the underlying mechanisms at play.

These two practical examples highlight the value of ShapeEmbed as a method to explore and discover shape variations in a fully unsupervised manner to investigate, untangle, and understand complex shape variations in biological experiments. Having methods that allow such an unbiased exploration of the distribution of biological shapes can be valuable in many settings, from assessing the efficacy of drug treatments to analyzing biopsies, where cell type identification must ideally be carried out without prior knowledge or potentially biased manual annotation.

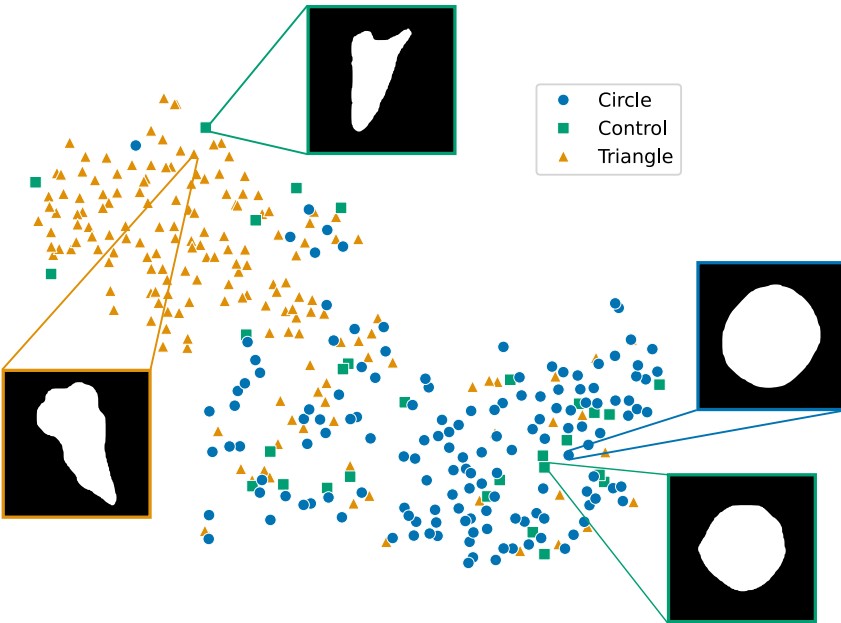

Figure 10: Projection (t-SNE) of the MEF latent space learned by ShapeEmbed. Data points are grouped according to the shape of the micropatterned Fibronectin-coated surface on which cells have been grown how they have been micropatterned. Cells in the control class have been seeded on a surface devoid of any micropattern. The control cells appear to adopt heterogeneous shapes that cover the range of morphologies exhibited by cells grown on both the triangle- and circle-patterned surfaces.

## G.3   Robustness to Segmentation Quality

The results presented in the main manuscript rely on clean segmentation masks as ShapeEmbed aims to learn a representation of shape information. Although image segmentation has received comparatively much more attention in the microscopy image analysis literature than shape representation learning [Lucas et al., 2021], it remains a challenging problem. To reassure about the usefulness of ShapeEmbed in real-world applications where clean segmentation masks may be difficult to obtain, we demonstrate in the following that ShapeEmbed is able to accurately encode shape information even when poor image quality results in poor segmentation quality.

To carry out these experiments, we synthetically degraded the MEF dataset with various levels of image noise, simulating a combination of Poisson shot noise and Gaussian readout noise as encountered in microscopy image acquisition. Given a ground truth image $I$, we first normalize it to the $[0, 1]$ range and denote the normalized image as $I_{\text{norm}}$. To account for the shot noise, we then scale this image by a factor $p \in \{1, 10, 100, 1000\}$ and sample each pixel value from a Poisson distribution using $\lambda = p \cdot I_{\text{norm}}$, yielding $I_{\text{P}}$. To account for the readout noise, we add pixel-independent Gaussian noise with standard deviation $g$, resulting in the degraded image $I_{\text{PG}}$. To reverse the intensity scaling and make a comparison to $I_{\text{norm}}$ meaningful, we divide $I_{\text{PG}}$ by $p$, resulting in the final degraded image $I_{\text{noisy}} = I_{\text{PG}}/p$. To evaluate the level of degradation of the image, we compute the peak signal-to-noise ratio (PSNR) between $I_{\text{noisy}}$ and the normalized ground truth image $I_{\text{norm}}$. This procedure allows us to control the noise severity via the scaling factor $p$ and construct four progressively degraded datasets corresponding to $p = 10000, 100, 10, 1$, which we refer to as "low", "mid-low", "mid-high". and "high" noise levels, respectively. In this setup, changing the scaling factor $p$ corresponds to recording a microscopy image with different exposure or laser power to control the amount of available light.

We then segment individual object instances in the degraded MEF dataset with high, mid-high, mid-low, and low noise levels using the state-of-the-art microscopy image segmentation deep learning algorithm cellpose [Stringer et al., 2021], keeping all parameters at their default value. From the instance segmentations obtained with cellpose, we create binary masks for each individual objects and extract object contours using the `find_contours` function from the scikit-image library

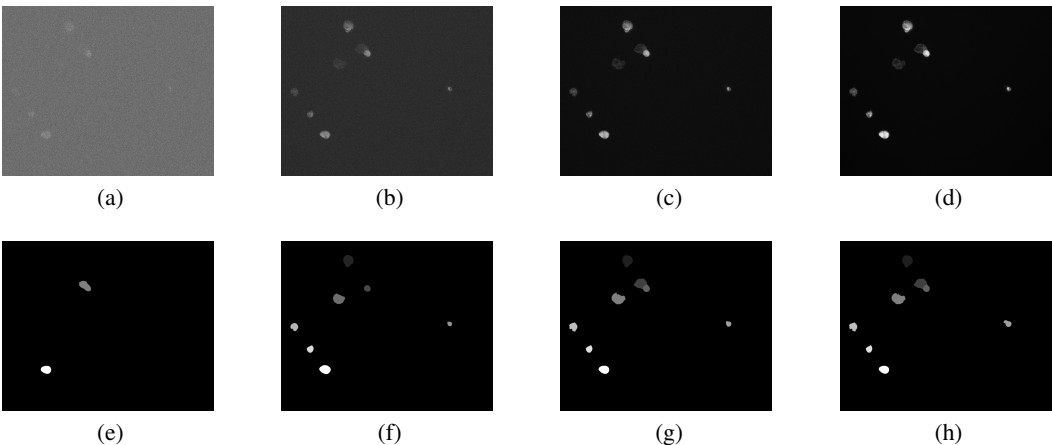

Figure 11: Sample images from the "circle" class of the MEF dataset illustrating the various noise levels (top row) and the resulting instance segmentations (bottom row) obtained with cellpose. High (a and e), mid-high (b and f), mid-low (c and g), and low (d and h) noise. Examples of original, noise-free images from the MEF dataset can be found in Figure 2.

(`https://scikit-image.org/`, MIT license). Each contour is resampled to a fixed number of points (64 for the MEF dataset) using the `splprep` and `splev` spline interpolation functions of the scipy library (Virtanen et al. [2020], BSD-3-Clause license). We finally compute Euclidean distance matrices from the resampled contour points to input them to ShapeEmbed.

We provide illustrative images of the different noise levels and of the resulting segmentation results in Figure 11, and a summary of the dataset we generate in Table 14.

Table 14: Summary of the noise-degraded versions of the MEF dataset.

| DATASET | POISSON SCALING $p$ | GAUSSIAN STANDARD DEVIATION $g$ | PSNR | NUMBER OF OBJECTS | IOU |
|---------|---------|---------|------|------|-----|
| HIGH | 1 | 0.9 | 0.345 | 2071 | 0.082 |
| MID-HIGH | 10 | 0.9 | 17.249 | 17026 | 0.649 |
| MID-LOW | 100 | 0.9 | 29.536 | 19299 | 0.698 |
| LOW | 1000 | 0.9 | 39.860 | 19762 | 0.702 |
| NONE (GT) | NONE | NONE | NONE | 16381 | 1 |

We measure the performance of ShapeEmbed on a downstream classification task as described in Section 4.2 of the main manuscript and report the F1-score as metric. In 12, we report the mean and standard deviation of the classification results on the four considered noise conditions (high, mid-high, mid-low, and low), as well as on the original, non-degraded dataset. We observe that even though performance increases as the noise level decreases as we would expect, ShapeEmbed performs well (F1-score$\leq 0.7$) as soon as segmentation quality is decent, which already happens in the mid-low noise level. Interestingly, ShapeEmbed achieves excellent performance even in the highest level of noise as, in this condition, the images are so bad (PSNR $\approx 0$) that only a very small subset of objects (2071 out of 16381, amounting to 12%) are segmented and systematic class-specific segmentation errors make it straightforward to distinguish different classes - resulting in a comparatively simpler classification problem. The reported performance in high noise conditions is therefore more reflecting of segmentation artefacts than of ShapeEmbed's performance. We nevertheless decided to include it for the sake of completeness.

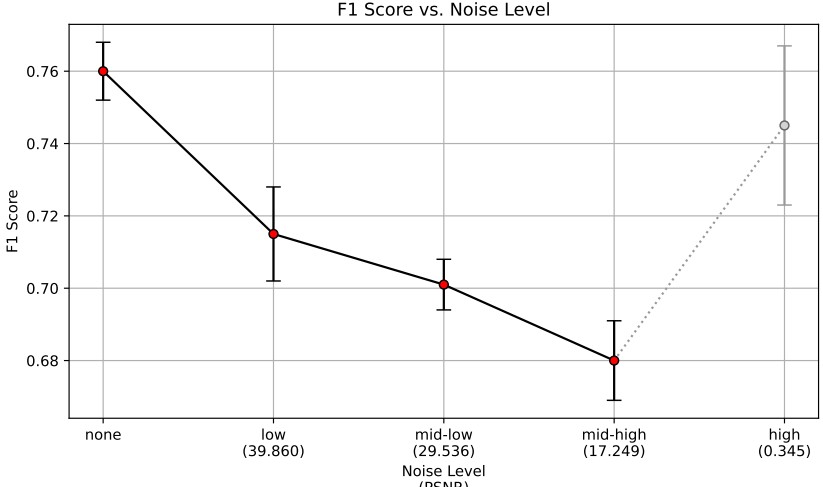

Figure 12: F1-score across varying levels of noise levels degrading the MEF dataset. The x-axis reports qualitative noise levels corresponding to increasing Poisson scaling factors (none to high), while the y-axis reports the mean F1-score across the MEF dataset degraded with varying noise levels. Red dots correspond to the mean F1-score, with black lines indicating the standard deviation.

### G.4  Generative Properties

ShapeEmbed's generative properties are particularly interesting for bioimaging applications, as they make it possible to qualitatively explore the shape distribution present in the dataset by calculating statistics or carrying out sampling in the latent space. We illustrate in Figure 13 three examples of contours randomly-picked among the data points of the two classes of the BBBC010 dataset, along with the mean outline of each of these classes (generated as in F.3). While the first three outlines illustrated in each row of Figure 13 are the contours of objects that were present in the original BBBC010 dataset, the mean outline does not correspond to a data point and is obtained thanks to the generative properties and decoding capabilities of our model. We observe that mean outlines accurately capture the shape signatures expected from each of the classes: straight for the dead class, and curved for the live class.

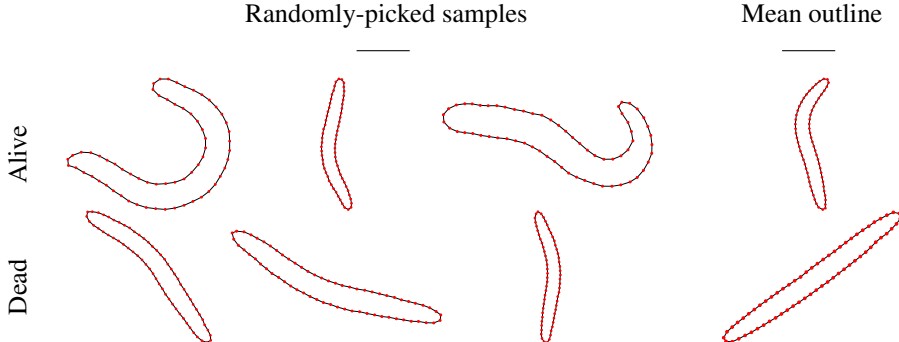

Figure 13: Mean outline generation in the BBBC010 dataset. We illustrate 3 data points randomly picked among the two classes of BBBC010 dataset (first three columns), along with the mean outline obtained by decoding the vector corresponding to the centroid of each of the classes in the ShapeEmbed latent space (last column).

We further depict in Figure 14 examples of outlines generated by randomly sampling the latent space learned by ShapeEmbed on BBBC010. The vectors were sampled and decoded into outlines as in F.3. We observe that the newly-generated outlines, which do not correspond to any "real" data point, are similar to the shapes present in the original BBBC010 dataset and look like plausible *C. elegans*

nematode outlines. This kind of generative capability can be of strong interest for the shape-aware generation of synthetic data as well as for hypothesis testing.

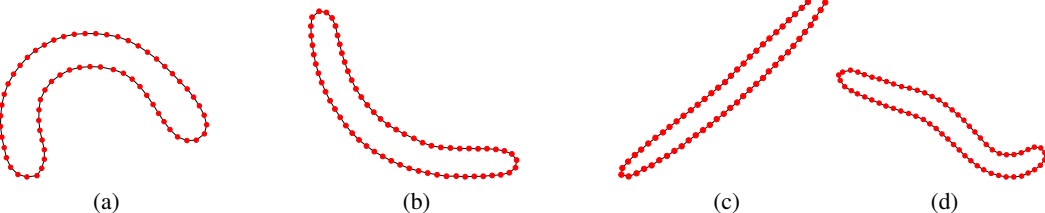

|        (a)        |        (b)        |        (c)        |        (d)        |

Figure 14: Novel outline generation in the BBBC010 dataset. We illustrate 4 outlines obtained by randomly sampling vectors from a normal distribution in the ShapeEmbed latent space that are subsequently reconstructed through the decoder path.

## H    Application to open curves

While ShapeEmbed is built for closed contours, it can in principle be applied to open curves as well, as the distance matrix representation does not require a closed contour. In fact, applying the approach to open contours simplifies the problem, as it leaves only two possible starting points to index the contour. As a result, the circular padding we implemented in the encoder is no longer required and the indexation-invariant loss can be simplified to only consider two possible indexations instead of $2N$ for a contour composed of $N$ points.

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
