# OpenReview forum: "ShapeEmbed: a self-supervised learning framework for 2D contour quantification"
_NeurIPS.cc/2025/Conference — NeurIPS 2025 poster_

### Official Review · Reviewer_pLNK · 2025-06-26

**Clarity:** 4
**Significance:** 3
**Originality:** 3
**Rating:** 5
**Confidence:** 3

**Summary:**

This manuscript presents ShapeEmbed, a self-supervised framework for learning transformation-invariant shape descriptors from segmentation masks. The use of pairwise distance matrices, architectural symmetry, and indexation-aware loss design is well-motivated and results in both strong discriminative performance and generative capability. Extensive experiments support the method’s utility across multiple datasets. The approach is timely and well-suited for bioimaging applications.

**Questions:**

--This paper discuss the case of closed curve, how the proposed descriptor describe the open curve?
-- When there are two many points, the scale of the distance matrix will be very large, how about the  efficiency of the proposed algorithm?
--Given the same 2D contour, when we sampled it with different number of points, for example, one for 100 and the other for 1000, then we will achieved two different distance matrix and different latent features, how about the similarity of these two latent feature? That is, the proposed representation is also invariant to the dense of the sampled points or not?
--Can this method can be generalized to the case of 3D space curve or not?

**Ethical Concerns:**

["NO or VERY MINOR ethics concerns only"]

**Limitations:**

--When there are a large number of the sampled points, the efficiency  of the proposed method may very poor.
--The invariance claims (translation, rotation, scaling, reflection, indexation) are central to the novelty of ShapeEmbed. While most are intuitively justified via the properties of distance matrices, the reflection and indexation invariance depend on architectural choices and loss design, which require clearer theoretical treatment.

**Paper Formatting Concerns:**

I have not find any paper formatting concerns.

**Quality:**

3

**Strengths And Weaknesses:**

---Strengths: This paper presents a technically sound and well-motivated framework for learning transformation-invariant shape representations using distance matrices and a customized VAE architecture.  The approach is novel, particularly in how it handles reflection and indexation invariance—two aspects often neglected in prior work.  The experimental evaluation is thorough, covering multiple natural and biological datasets, and is supported by rich ablation studies, visualizations, and generative sampling.  The method is clearly relevant for biological imaging and other applications requiring shape-based analysis.  The manuscript is clearly written, well-structured, and supported by detailed supplementary materials, which enhance the reproducibility and transparency.

---Weaknesses: Despite the comprehensive classification experiments, the paper falls short in demonstrating the broader applicability of the learned shape representations. Additionally, on biological datasets like MEF and BBBC010, the Region Prop. method performs competitively, suggesting that much of the discriminative power may stem from simple size or area cues. The paper would benefit from a deeper analysis of whether ShapeEmbed captures richer, non-size-based shape information, particularly in biologically relevant contexts.

---

> ### Author Rebuttal · Authors · 2025-07-30
>
> ## Applicability of shape representations and importance of size and area cues.
> The reviewer mentions that ShapeEmbed “falls short in demonstrating the broader applicability of the learned shape representations” and that  “much of the discriminative power may stem from simple size or area cues”. We are discussing these points in detail in our response to reviewer iSBr (“Added benefit of ShapeEmbed” and “Limited scope of the method”) and summarize the key elements here. Size indeed plays an important role in many biological imaging problems, which is why Region Properties perform competitively. That said, even in situations where size information is crucial, region properties only capture limited information about shape and are therefore outperformed by the scale-invariant version of ShapeEmbed on all the biological imaging datasets we considered except MEF, as illustrated by the new experiments we present in our reply to reviewer VYfD (7% better on MOC and 23% better on HeLa Kyoto). Importantly, adding size information back to the ShapeEmbed latent codes only yields an additional improvement of 2% and 5%, respectively. We believe all-in-all our results highlight the power of learned shape descriptors beyond simple size and area cues in biological imaging data, as well as in more classical imaging data (see Table 1 in our manuscript).
>
> ## Application of the method to open curves.
> We did not explicitly investigate open curves in our manuscript as our target application is biological imaging, where objects are most often described by their closed contour. That said, ShapeEmbed can be applied to open curves as well. In fact, open curves are simpler to deal with than closed ones as they only have two possible starting points: the sole remaining ambiguity is the direction in which the points are traversed. As a result, the circular padding we implemented in the encoder would no longer be required and our indexation-invariant loss would be simplified to only consider 2 possible indexations instead of 2N for a contour composed of N points. We propose to include this point in a new section of our Supplementary Material as follows:
>
> ***Application to open curves**. While ShapeEmbed is built for closed contours, it can in principle be applied to open curves as well, as the distance matrix representation does not require a closed contour. In fact, applying the approach to open contours simplifies the problem, as it leaves only two possible starting points to index the contour. As a result, the circular padding we implemented in the encoder is no longer required and the indexation-invariant loss can be simplified to only consider two possible indexations instead of 2N for a contour composed of N points.*
>
> ## Impact of the number of points on the efficiency of the method.
> It is true that the number of points used to create the distance matrix will influence the amount of required memory as well as the required training time. We invite the reviewer to read our response to reviewer iSBr for a detailed experimental analysis of this topic.
>
> ## Invariance with respect to the number of sampled contour points.
> As described in Section 3.1 of our manuscript, we draw points from an object’s contour to create a distance matrix by first interpolating the pixels originally composing the object outline, and then uniformly sampling a given number of points from that interpolation. As pointed out by the reviewer, it is therefore possible to sample the same 2D contour with different numbers of points. We can expect each contour to possess an optimal sampling threshold that defines the minimum number of points required to accurately represent it. Insufficient sampling below this threshold results in an inadequate contour description, while oversampling beyond this threshold does not increase description accuracy. This minimum number of points correlates directly with geometric complexity: smooth contours have lower sampling requirements, whereas complex, irregular contours necessitate denser sampling. The notion of invariance to sampling pointed out by the reviewer would really only apply to oversampled contours, as undersampled contours would result in different shapes and would therefore, by design, not have the same representation in our latent space. As it is highly challenging to know a priori what the minimum number of points for a given contour is, it is extremely hard to formulate a model that would implement invariance to sampling only when appropriate. In our current architecture, we therefore implemented the number of points used to describe the contour as a hyper parameter that must be fixed prior to training.
>
> In the future, it could be possible to build models with variable point numbers to have a framework ready to implement sampling invariance. Supporting variable numbers of points would also allow for the training of inhomogeneous datasets. One further area of interest for future work is the design of a flexible method that could add or remove points by encoding and decoding a matrix allowing us to increase or decrease the level of detail.
>
> ## Application of the method to 3D curves.
> We are confident that our method can be adapted to curves in 3D space as the distance matrix representation as well as the MDS reconstruction algorithm also apply in higher dimensions [1]. In preliminary follow-up work to this manuscript, we successfully applied our method to synthetic curves in 3D space. We however believe that a more exciting future direction is to adapt our method to 3D surfaces described as point clouds. The main obstacle towards that goal is that, unlike points on a contour, points on a surface do not have a clear order. This in turn requires a new approach to achieving indexation invariance, as considering all shifted versions of the distance matrix is no longer sufficient in such a setting.
>
> [1]: Dokmanic, Ivan, et al. "Euclidean distance matrices: essential theory, algorithms, and applications." IEEE Signal Processing Magazine 32.6 (2015): 12-30.

---

> > ### Author Response · Authors · 2025-08-07
> >
> > We thank the reviewer for their time and insights into our work. We hope that our response has clarified the points raised, and we would welcome any further feedback or comments.

---

### Official Review · Reviewer_iSBr · 2025-06-29

**Clarity:** 4
**Significance:** 3
**Originality:** 2
**Rating:** 5
**Confidence:** 3

**Summary:**

The paper introduces ShapeEmbed, a self-supervised learning (SSL) framework that is invariant to scaling, translation, rotation, reflection, and re-indexing of shape contours. The input is the cross-distance matrix between all points of the input contour. The model is based on a VAE architecture (which includes circular padding in its convolution layers), which, together with a novel loss function, allows for invariance to indexation. The method seems novel, and the experiments (along with ones in the supplementary) provide strong support for the authors’ claims.

**Questions:**

This is based on the Weaknesses section, questions are ordered by importance:
1. As mentioned in the Weaknesses section, my main concern is that the method is not scalable. What is the trade-off between the number of contour points and training time (for at least some of the datasets)?

2. What is the number of trainable parameters of the model? ResNet18 usually has ~11M parameters. Does that mean ShapeEmbed (Enc. + Dec.) has ~22M? If so, this might explain the long training time. Can a simple CNN architecture obtain comparable results in a shorter training time?

3. Computation: using 2N alternative versions, each matrix is entered twice, original and mirrored. - This might also explain the long training time and, what I imagine, high GPU RAM consumption. Is that notion correct? If so, can it be mitigated somehow?

4. Comparison with SSL methods (MAE and SimCLR) might not be fair - SimCLR training is known to be painfully long, did it properly converge? MAE, on the other hand, has 86M/304M/632M params. For B/L/H, respectively.
a) Training it on small datasets such as MEF/MPEG-7 (~1400 samples) might be ill-fitted.
b) The patch-based approach might not be adequate for small images (28x28 and 64x64 for MNIST and BBBC010). Have you considered comparing ShapeEmbed to the pretrained versions of SimCLR and/or MAE?

5. MNIST and MPEG are considered “easy” datasets. While on the harder ones (MEF, BBBC010) ShapeEmbed (w/o Sz) is not much
better than Region Prop. in L368-L370 you mention that “summary statistics, [...],  perform exceptionally well”. Can you elaborate on what the added benefits of ShapeEmbed?

6. Since the reconstruction seems optional, why not use a standard AE?

**Ethical Concerns:**

["NO or VERY MINOR ethics concerns only"]

**Final Justification:**

The authors have addressed my concerns in their detailed rebuttal. They have performed additional experiments w.r.t the proposed method's scalability and the fairness of their comparison with the competing method. After reviewing the rebuttal, as well as the rest of the reviews and associated rebuttals, I have decided to increase my overall score.

**Limitations:**

The limitations appear as part of the "NeurIPS Paper Checklist".

**Paper Formatting Concerns:**

I did not notice any formatting issues.

**Quality:**

3

**Strengths And Weaknesses:**

# Strengths:
1. The paper is well written. The introduction is clear, and the problem formulation and motivation are well defined. The paper is clearly placed w.r.t. the related literature.
2. The authors break down different invariances and thoroughly explain how ShapeEmbed obtains them:
* Distance Matrix - translation, rotation and scale invariance.
* Process mirrored version of the distance matrix - invariance to the direction of travel.
* Circular padding - indexation invariance.
3. Experiments - the experiments are divided into three:
* Scaling and Indexation Invariance.
* Rotation and Translation Invariance.
* Application to Biological Imaging.
with the addition of an ablation study.

The results suggest the proposed method indeed encodes the aforementioned invariances in its architecture and training procedure. This is evident when ShapeEmbed is compared to standard baselines and the main competitors.

4. Supplementary Material (SUPMAT) - There is excellent information in the SUPMAT. Additional experiments, ablations and visualization. I recommend that the authors consider including figure 5 (mean outline reconstruction) and figure 7 (t-sne for the BBBC010 dataset). I understand that this might not be possible due to space limitations.

# Weaknesses
## Major Weaknesses and Questions:
1. Scalability and runtime - my main concern is that the method is not scalable. As mentioned in SUPMAT Section C. L48-50, it takes ~49 hours to train the model on MNIST with 64 points per image contour (or 32 according to Table 1. in the SUPMAT?) on a A100 GPU. I would like to see the trade-off between the number of contour points and training time to performance for at least some of the datasets.

2. What is the number of trainable parameters of the model? ResNet18 usually has ~11M parameters. Does that mean ShapeEmbed (Enc. + Dec.) has ~22M? If so, this might explain the long training time. Can a simple CNN architecture obtain comparable results in a shorter training time?

3. Indexation Invariant Reconstruction Loss: by using 2N alternative versions, each matrix is entered twice, original and mirrored. - This might also explain the long training time and, what I imagine, high GPU RAM consumption.

4. Comparison with SSL methods (MAE and SimCLR) might not be fair - SimCLR training is known to be painfully long, did it properly converge? MAE, on the other hand, has 86M/304M/632M params. for B/L/H, respectively. Training it on
small datasets such as MEF/MPEG-7 (~1400 samples) might be ill-fitted. Additionally, the patch-based approach might not be adequate for small images (28x28 and 64x64 for MNIST and BBBC010). Have you considered comparing ShapeEmbed to the pretrained versions of SimCLR and/or MAE?

## Minor Weaknesses
1. Limited scope - the paper tackles a problem that might be considered `limited' - invariance to geometric properties of well-defined shape contours. That being said, I can see how the method can be applied in a more extensive scope and on top of foundation models such as Segment Anything Model (SAM).
2. MNIST and MPEG are considered “easy” datasets, while on the harder ones (MEF, BBBC010) ShapeEmbed (w/o Sz) is not much
better than Region Prop. in L368-L370 you mention that “summary statistics, [...],  perform exceptionally well”. Can you elaborate on what the added benefits of ShapeEmbed?

3. Since the reconstruction seems optional, why not use a standard AE?

## Typos:
L251 - but not to to the same

---

> ### Author Rebuttal · Authors · 2025-07-31
>
> ## Scalability of the method and impact of the number of contour points on training time.
> To illustrate the impact of the number of contours points on training time, we report below the time required to complete one epoch for each of the considered dataset and with increasing number of contour points in the input distance matrices. The reported times are in seconds and calculated as the average time for a single epoch of training over 400 epochs. In parenthesis, we also indicate the total time for the full 400 epoch training. We ran these experiments on a GPU cluster with NVIDIA A100 GPUs. While the number of contour points indeed has an impact on training time, it is dominated by dataset size as one epoch requires processing the entire dataset once. More specifically, an increase in the number of points by a factor of 8 (32 points to 256) only leads to an increase by a factor of 1.77 (175s to 311s) in training time per epoch.
>
> ||32|64|128|256|
> |---|---|---|---|---|
> |MNIST (70’000 images)|174.5862s (~19h24)|180.5762s (~20h06)|270.49s (~30h03)|310.96s (~35h04)|
> |MEFS(1’407 images)|68.4477s (~7h36)|70.1835s (~7h48)|74.2795s (~8h15)|86.1598s (~9h34)|
> |BBBC010(26’198 images)|6.1271s (~41min)|6.1887s(~41min)|6.6136s (~44min)|7.5953s (~51min)|
> |MPEG-7(1’400 images)|4.99s (~33min)|5.17s (~34min)|5.37s (~36min)|6.32s (~42min)|
>
> ## Number of trainable parameters and comparison to a simple CNN
> We report below the number of parameters in ShapeEmbed based on the input size. To obtain these values, we used the ‘summary’ functionality of the ‘torchinfo’ library. Variations in the number of contours points, and therefore in the input size, have a minimal impact on the total number of parameters.
>
> |Input size|Total number of parameters|
> |---| --- |
> |32x32|14153889|
> |64x64|14161841|
> |128x128|14163769|
> |256x256|14164221|
>
> The reviewer also asks whether a simple CNN could obtain comparable results in a shorter training time. We are not sure if “simple” refers to an architecture that would not be based on distance matrices (instead on images), or to a smaller network architecture (instead of a ResNet backbone). An architecture that would not be based on distance matrices would not benefit from built-in invariance properties, and would thus be substantially disadvantaged as compared to our method, as seen by the performance of O2VAE. A lighter version of our encoder and decoder architectures could certainly be conceived, but would require extensive ablation experiments to validate it over a well-established backbone like ResNet. We therefore believe this to be a valuable exploration for future work, but to go beyond the scope of our current paper.
>
> ## Efficiency of the indexation invariance implementation
> It is true that the encoder processes each matrix twice to achieve reflection invariance. However, our loss function only compares the output of the decoder to 2N versions of our target matrix, which are computed on the fly, meaning that the decoder doesn’t have to run 2N times. In the future, we could make the encoder invariant to reflection without requiring it to be executed twice by using reflection equivariant convolutions [1]. We propose to mention this in our conclusion section as follows:
> *"While our current encoder achieves reflection invariance by processing each matrix as well as as a flipped version, this computational overload could be avoided in the future by using reflection equivariant convolutions [1]."*
>
> [1]: Cohen, Taco, and Max Welling. "Group equivariant convolutional networks." International conference on machine learning. PMLR, 2016.
>
> ## Fairness of the comparison with standard SSL methods
> The MAE and SimCLR models we trained converged for all datasets in our original experiments. We confirmed this by re-running the experiments to answer this comment. We unfortunately cannot provide figures of the loss curves in the rebuttal, but could add them to our Supplementary Material.
>
> The reviewer also asks for a comparison of ShapeEmbed to pretrained versions of SimCLR and/or MAE. Below, we report the F1-score of the downstream classification tasks of each of the datasets (including two additional biological imaging datasets, see our response to reviewer VYfD) using pretrained MAE models. To do these experiments, we initialized our MAE model using the pytorch checkpoints provided in the original Github repository from Facebook Research. We did not include results with a pretrained SimCLR model as the implementation we use (link to the repo in the original manuscript) neither provides pretrained weights nor a straightforward way to initialise the model for inference. As the other pretrained SimCLR models we could find in platforms like Hugging Face would require major changes to fit in our pipeline, we were unable to experiment with them during the short time of the rebuttal.
>
> Performance increases across all MAE modalities when using pretrained models. We believe that this can be attributed to the large number of images the pretrained models have already been exposed to, which help compensate for the relatively small amount of data available in all datasets except MNIST. ShapeEmbed, trained from scratch, however continues to outperform these competing methods. It further suggests that the invariances built into ShapeEmbed allows it to learn meaningful representations even with limited data. We thank the reviewer for suggesting this experiment and will include these results in Table 1 in the final version.
>
> **Benchmark datasets**
> ||MNIST|MPEG-7|
> |---|---|---|
> |MAE ViT-B pretrained|0.92±0.01|0.42±0.01|
> |MAE ViT-L pretrained|0.93±0.01|0.51±0.02|
> |MAE ViT-H pretrained|0.94±0.01|0.57±0.02|
> |ShapeEmbed|0.96±0.01|0.75±0.02|
>
> **Biological imaging datasets (previous)**
> ||MEF|BBBC010|
> |---|---|---|
> |MAE ViT-B pretrained|0.61±0.01|0.65±0.08|
> |MAE ViT-L pretrained|0.64±0.02|0.77±0.03|
> |MAE ViT-H pretrained|0.65±0.01|0.76±0.06|
> |ShapeEmbed|0.67±0.01|0.837±0.01|
> |ShapeEmbed+size|0.76±0.01|0.87±0.01|
>
> **Biological imaging datasets (new)**
> ||MOC|HeLa Kyoto|
> |---|---|---|
> |MAE ViT-B pretrained|0.48±0.15|0.72±0.02|
> |MAE ViT-L pretrained|0.66±0.03|0.76±0.03|
> |MAE ViT-H pretrained|0.63±0.05|0.73±0.08|
> |ShapeEmbed|0.68±0.01|0.80±0.04|
> |ShapeEmbed+size|0.70±0.05|0.85±0.04|
>
> ## Limited scope of the method
> The reviewer points out that the problem tackled by ShapeEmbed may be “considered limited”. While we agree that shape quantification has not gotten as much attention as other problems such as image segmentation, we believe it to be an important challenge in particular in biological imaging. We therefore consider that it is relevant to propose innovative methods focusing on the quantification of segmented images, especially with biological imaging in mind.
>
> We fully agree with the reviewer that our method can be combined with upstream segmentation methods such as SAM, and have ourselves combined it with cellpose (see Supplementary Section G.3).
>
> ## Added benefit of ShapeEmbed
> We respectfully disagree with the characterization of MPEG-7 as an “easy” dataset, in that its small size (1400 images for 70 classes) makes it highly challenging for DL methods which typically require many more images to generalize well. Most DL models struggle to learn meaningful representations on MPEG-7, as demonstrated in our experiments. In contrast, ShapeEmbed performs despite the large number of classes and the small number of images per class in this dataset, which speaks to its efficiency and suitability in low-data regimes.
>
> The reviewer also requests an explanation of why ShapeEmbed without size does not exhibit much better performance than summary statistics (region properties) on the biological imaging datasets. In the context of biological imaging, size can be relevant to the phenomenon observed. Several of the classical summary statistics (e.g., area and perimeter) explicitly encode size while ShapeEmbed encodes “shape” as traditionally-defined, without size information [1]. As a result, ShapeEmbed without size is disadvantaged in experiments where size is an essential feature for the downstream classification task. Region properties are however inherently limited in capturing shape variability as they reduce contours to a small set of global statistics and fail to capture localized variations. This is illustrated by the fact that, when size is explicitly added back to the ShapeEmbed latent codes, our method not only becomes competitive but surpasses region properties. Unlike region properties, ShapeEmbed in addition learns a continuous and structured latent space where shapes that cluster together are semantically and morphologically similar. This geometric interpretability is hard to build into region properties and similar handcrafted approaches. Furthermore, our method is generative and thus enables visual investigation of the latent space as well as synthesis of novel shapes, as illustrated in Supplementary Section G.1, Figures 5 and 6. As a result, ShapeEmbed is not only a classification tool but also enables exploratory shape analysis.
>
> We also point to our reply to reviewer VYfD, where we further discuss MNIST and MPEG-7 and provide results on two additional biological imaging datasets.
>
> [1] I.L. Dryden & K.V. Mardia (1998). Statistical Shape Analysis. John Wiley & Sons. ISBN 978-0-471-95816-1
>
> ## Use of a standard AE instead of a VAE
> Our method could indeed in principle use a standard AE architecture. However, the VAE provides a dense continuous latent space enabling interpolation and averaging as illustrated in Supplementary Figures 5 and 11. This capability also makes it possible to generate shapes de novo according to the learned distribution, as shown in Supplementary Figures 6 and 12. While we did not quantitatively explore these capabilities in depth in our manuscript as it focuses on the construction of the model itself, they are of strong interest to biological imaging and are therefore an essential feature of our design.

---

> ### Comment · Reviewer_iSBr · 2025-08-03
>
> The authors have addressed my concerns in their detailed rebuttal. They have performed additional experiments w.r.t the proposed method's scalability and the fairness of their comparison with the competing methods, with encouraging results. After reviewing the rebuttal, as well as the rest of the reviews and associated rebuttals, I have decided to increase my overall score.

---

### Official Review · Reviewer_REU4 · 2025-07-01

**Clarity:** 2
**Significance:** 2
**Originality:** 2
**Rating:** 4
**Confidence:** 3

**Summary:**

The paper is about self-supervised learning for 2D contours. The authors aim to inject translation, scaling, rotation, reflection invariance to the representation. So this can work for the specific 2D contour task.

**Questions:**

No.

**Ethical Concerns:**

["NO or VERY MINOR ethics concerns only"]

**Final Justification:**

Thanks for the response. Some of my concerns are addressed. I keep my current rating because I still believe the application and the dataset is somewhat limited.

**Limitations:**

See the previous section. While the paper showed some interesting results, its generalization capabilities raise concerns – particularly regarding large-scale datasets and broader problem applicability

**Quality:**

2

**Strengths And Weaknesses:**

I think the paper delivered some interesting and strong results. To make the representation invariant to several transformations, the authors first used the distance matrices as the inputs and then carefully modified conv-based networks and loss functions. In the end, the representation is invariant to translation, scale, rotation and reflection.

However I have the following concerns,
1. The core idea is basically incorporating inductive bias into the design. While this enables native handling of the target problem, it may compromise generalization capabilities. This trade-off explains the results in Table 4: performance improves when size information is included, yet other tables demonstrate that such information isn't universally required.
2. The experiments are all about small datasets such as minist.

---

> ### Author Rebuttal · Authors · 2025-07-30
>
> ## Inductive bias and generalization capability.
> The reviewer’s comment is as follows:
>
> *“The core idea is basically incorporating inductive bias into the design. While this enables native handling of the target problem, it may compromise generalization capabilities. This trade-off explains the results in Table 4: performance improves when size information is included, yet other tables demonstrate that such information isn't universally required.”*
>
> The explicit incorporation of rotation invariance, scaling invariance, translation invariance, reflection invariance, and indexation invariance (often referred to as parameterization invariance in the shape analysis literature) is indeed at the core of our method. While these invariances may be considered to be inductive biases, they are essential in characterizing shape as classically defined to be the information that remains regardless of these geometric transformations [1]. Generally speaking, building in the correct inductive biases for a given problem increases generalizability, prevents overfitting, and reduces the amount of required training data [2].
>
> We however recognize that invariance to scaling, although included in the classical definition of shape, is not always necessary and can even be counter productive depending on the application. In biological imaging specifically, the size of objects (e.g., cells, organisms) often carries important information. To find a middle ground that both acknowledges the classical definition of shape and remains flexible in situation where size matters, we therefore chose to leave the option to reintroduce scale information by concatenating it to the latent vector for downstream tasks when desired, as demonstrated in our experiments by the “ShapeEmbed” and “ShapeEmbed + size” versions. We believe that the suitability of a scale-invariant model genuinely depends on the application and is therefore a choice better left to the user. To clarify these points, we propose to expand the discussion in Section 4, line 359 in the final version of the paper:
>
> *While scale invariance is traditionally considered to be essential for shape analysis [1] and our framework is inherently scale-invariant, scale may be a crucial feature in some applications. To account for this, we additionally consider a variant of ShapeEmbed that preserves scale information by saving the norm of the distance matrix before normalization and concatenating it to the latent code for downstream classification. The decision to consider scale as a relevant feature to be included or as a nuisance transformation to be removed entirely depends on the use case and biological question considered.*
>
> [1]: Klingenberg, Christian Peter. "Walking on Kendall’s shape space: understanding shape spaces and their coordinate systems." Evolutionary Biology 47.4 (2020): 334-352.
>
> [2]: Zhang, Chiyuan, et al. "Identity Crisis: Memorization and Generalization Under Extreme Overparameterization." International Conference on Learning Representations. (2019)
>
>
> ## Size of the datasets.
> The reviewer states that all the experiments are performed on small datasets and specifically raises concerns about ShapeEmbed’s ability to scale to larger datasets. As we are not completely sure what the reviewer specifically means by “small” datasets, we address two possible interpretations of this concern. We would also like to point out that we have now run new experiments on additional biological imaging datasets in our reply to reviewer VYfD.
> Should the concern be with ShapeEmbed’s ability to handle datasets that are composed of many images, our experiments are performed on datasets of various sizes as indicated in Table 1 of Supplementary section C “Additional Details on the Considered Datasets”. In our two computer vision benchmarks, the MPEG-7 dataset is representative of a small dataset (1400 objects) while MNIST is representative of a large dataset (70’000 objects). Similarly for the biological imaging datasets, MEF is representative of a small dataset (1’407 objects) while BBBC010 is representative of a large dataset (26’198 objects).
>
> If the concern instead has to do with the size of the original images (from which contours are extracted), we recall that ShapeEmbed takes distance matrices built from points composing the contour of single objects as input. As the points of the contour are sampled from a continuous spline interpolation of the original pixels forming the outline of the object, the number of points we draw is a free parameter that is independent from the resolution of the original image. For instance, we draw 32 points to describe the contour of objects in the 28x28 images of the MNIST dataset, and 64 points to describe the contour of objects in the 128x128 images  of the MPEG-7 dataset. The number of points used to describe the contour is a free parameter that is set empirically depending on the size of the objects in the dataset considered and the observed performance (as having too few points obviously results in poorly described shapes and therefore a less performant model), but we do not observe it to significantly impact training time. For more information, we point the reviewer to our answer to reviewer iSBr, we further demonstrate the impact of the choice of the number of contour points (and therefore the size of the resulting distance matrix) on training time and number of parameters in the model.

---

> > ### Author Response · Authors · 2025-08-07
> >
> > We thank the reviewer for their time and insights into our work. We hope that our response has clarified the points raised, and we would welcome any further feedback or comments.

---

> > ### Comment · Area_Chair_F1Li · 2025-08-07
> > **plz provide feedback**
> >
> > Dear REU4 (including also the authors in this one),
> >
> > Thank you for your review. Could you please check the authors' rebuttal and provide feedback?
> >
> > Thank you AC

---

### Official Review · Reviewer_VYfD · 2025-07-02

**Clarity:** 4
**Significance:** 2
**Originality:** 2
**Rating:** 3
**Confidence:** 4

**Summary:**

This paper investigates the problem of accurately describing object contours in 2D images and proposes a novel method called ShapeEmbed. Based on a learning-based framework, ShapeEmbed learns a shape representation that is invariant to translation, scaling, rotation, reflection, and point indexing, and applies it to downstream tasks. Experiments are conducted to validate the effectiveness of the proposed method.

**Questions:**

Please refer to the weaknesses section, and in particular, highlight the originality of your work—rather than presenting it as an incremental contribution (i.e., a mere combination of existing modules). Emphasize whether your approach addresses a previously unsolved challenge.

**Ethical Concerns:**

["NO or VERY MINOR ethics concerns only"]

**Final Justification:**

The authors do not address my concerns w.r.t. novelty and contribution. I will keep my original score.

**Limitations:**

Yes

**Paper Formatting Concerns:**

No.

**Quality:**

2

**Strengths And Weaknesses:**

Strengths:

1. The paper is clearly written and easy to understand.

2. This paper focuses on a core challenge in object shape modeling, and correspondingly proposes a shape descriptor that is invariant to translation, scaling, rotation, reflection, and point indexing.

3. The effectiveness of the proposed shape descriptor is validated through experiments.

Weaknesses:

1. The novelty of the paper is not clearly articulated. The core challenges in shape representation—such as achieving invariance to translation, scaling, and rotation—have been long-standing problems addressed by many prior works. The use of a distance matrix to achieve these invariances is not particularly surprising, and similar approaches have been explored previously (as the authors themselves acknowledge). Then, what's the novelty and contributions?
If the key contribution lies in addressing the indexation invariance problem for closed contours, this should be clearly emphasized in the Introduction.

2. Has the issue of indexation invariance truly not been addressed in prior work? It is unclear whether the authors have conducted a sufficiently comprehensive review of related literature on this aspect.

3. There is a notable performance gap between the baselines on MNIST and MPEG-7. The authors should provide a clearer comparison of these two datasets, highlighting their differences in structure and task complexity during dataset description.

4. I'm not an expert in shape representation, but the number of datasets used in the experiments seems relatively limited. Are there additional benchmark datasets or comparison methods that could further support the generality and robustness of the proposed approach?

---

> ### Author Rebuttal · Authors · 2025-07-30
>
> ## Clarification of the novelty and contributions.
> As correctly stated by the reviewer and acknowledged in our manuscript, we are not the first to propose a distance matrix representation for shapes. However, to the best of our knowledge, we are the first to utilise distance matrices representing closed contours to derive a learned latent representation of shape enabling us to achieve superior accuracy in downstream classification tasks and opening the door to generative shape modelling (see Supplementary Figures 6 and 12). As the reviewer correctly states, one of our key contributions is to propose a novel solution to the problem of indexation invariance. In addition to the discussion we currently have on lines 58-67 of our manuscript, we propose to also state this more explicitly in the final version of our contribution list as follows:
> 1. We introduce, **to the best of our knowledge, the first self-supervised** representation learning model that learns shape descriptors from distance matrices. The descriptors are, by design, invariant to scaling, translation, rotation, reflection, and re-indexing.
> 2. We are, **to the best of our knowledge, the first** to propose a solution to achieve indexation invariance in a VAE architecture for shape description based on a padding operation in the encoder, operating jointly with a new loss function.
> 3. We show that our method outperforms the representation learning state-of-the-art and classical baselines on downstream shape classification tasks.
>
>
> ## Prior work on indexation invariance.
> After extensive literature research, we were unable to identify previous methods using distance matrices to derive a learned latent representation of shape, or methods that address indexation invariance with self-supervised learning models in the context of shape quantification. We would appreciate any hint to potentially missed prior work the reviewer is aware of, which we would be keen to include in the Related Work section.
>
> ## Performance difference between MNIST and MPEG.7.
> The reviewer points out the performance gap between the MNIST and MPEG-7 datasets and asks for clarifications as to what may be causing it. The information we provide in our Supplementary Material Section C entitled “Additional Details on the Considered Datasets” motivating the choice of the datasets we focus on and summarizing their properties (Supplementary Table 1) is of relevance to address this point. As indicated in Supplementary Table 1, MNIST is a very large (~70’000 images) and relatively easy dataset (thanks to its ~7’000 data points per class) that, even though not designed for shape analysis, serves as an excellent established deep learning model benchmark to assess the performance of ShapeEmbed against other existing methods. In contrast, MPEG-7 is a benchmark dataset designed for shape analysis, but not for deep learning models: it is comparatively much smaller than MNIST with only 1400 objects split between 70 classes with 20 objects each. In contrast to MNIST, MPEG-7 therefore allows us to assess the performance of ShapeEmbed and its competitors on limited, complex data composed of many categories for which only few instances are available (as is also often the case in biological imaging). We hypothesize that these differences in the properties of the two datasets are the reason for the much lower performance observed across all considered methods on MPEG-7. The small number of images per class in MPEG-7 makes this dataset particularly challenging for learning-based methods, while the classical summary statistics achieves respectable results. We consider it one of the strengths of ShapeEmbed that it is able to excel under such conditions.
>
>
> ## Additional datasets.
> In addition to the 4 datasets we already have included in our manuscript (two computer vision datasets, two biological imaging datasets), we propose to include two additional biological imaging datasets to further demonstrate the broad usability of ShapeEmbed. More precisely, the following text will be added to the Supplementary Material in Section C “Additional Details on the Considered Datasets” and the results will be included in Section G “Additional Biological Imaging Results” in the camera-ready version of the manuscript:
>
> *The HeLa Kyoto dataset (CellCognition project, 10.1038/nmeth.1486) consists of fluorescence microscopy images of H2B-mCherry-stained HeLa Kyoto cell nuclei, labeling chromatin and capturing nuclear morphology during mitosis. Already segmented and cropped masks for individual nuclei are available under the “classifier data” section of the dataset. We consider 4 classes that are representative of nuclei at key phases of mitosis (category name and number of samples in parentheses): early anaphase (earlyana, 96 nuclei), interphase (inter, 216 nuclei), metaphase (meta, 169 nuclei), and prometaphase (prometa, 168 nuclei).*
>
> *The MOC dataset (10.5281/zenodo.6554587) consists of fluorescence microscopy images of mouse osteosarcoma cells. In this dataset, cells have been exposed to cytoskeletal perturbation through treatment with the single drugs jasplakinolide (Jasp) and cytochalasin D (Cytd). The cells are divided in 3 classes: a control class with untreated cells (318 cells), a Cytd class (175 cells), and Jasp class (156 cells).*
>
> *In Table X (table number to be adjusted when incorporated in the document), we report the F1-scores obtained when classifying the HeLa Kyoto and MOC datasets with the competing methods considered in the main manuscript (Section 4.3) and ShapeEmbed.*
>
> |  | MOC | Hela Kyoto |
> | --- | --- | --- |
> | Region properties | 0.61 ±0.07 | 0.57 ±0.08 |
> | MAE ViT-B (not pretrained) | 0.57 ±0.02 | 0.47 ±0.13 |
> | MAE ViT-L (not pretrained) | 0.60 ±0.07 | 0.42 ±0.09 |
> | MAE ViT-H (not pretrained) | 0.51 ±0.09 | 0.44 ±0.21 |
> | SimCLR | 0.65 ±0.14 | 0.48 ±0.17 |
> | O2VAE | 0.60 ±0.06 | 0.63 ±0.08 |
> | ShapeEmbed | 0.68 ± 0.01 | 0.80 ± 0.04 |
> | ShapeEmbed + size | 0.70 ± 0.05 | 0.85 ± 0.04 |
>
> **Table X**. *Additional biological dataset results (F1-score ± σ, higher is better), for both classical methods and SotA learning.* methods.

---

> > ### Comment · Reviewer_VYfD · 2025-08-05
> >
> > Thank you for the response. Most of my concerns are addressed. But I still do not get your novelty.
> >
> > What do you mean by "we are not the first to propose a distance matrix representation for shapes. However, to the best of our knowledge, we are the first to utilise distance matrices representing closed contours to derive a learned latent representation of shape enabling us to achieve superior accuracy in downstream classification tasks and opening the door to generative shape modelling".
> >
> > So your novelty are:(1) utilise distance matrices representing closed contours? if so, what's the essential difference of representing closed contours? (2) enabling us to achieve superior accuracy in downstream classification tasks and opening the door to generative shape modelling. If so, this is a performance contribution?

---

> ### Author Response · Authors · 2025-08-05
> **ShapeEmbed's key contributions**
>
> Thank you for your reply. We are glad most of your concerns are now addressed and hope to provide clarifications on remaining questions about novelty.
>
> The key contribution of our work is using distances matrices together with a learned latent space using a Variational AutoEncoder. Distance matrices have been used to describe shapes[1] but not as input to a VAE to learn a latent representation that describes shapes as vectors allowing for interpolation averaging and generation as well as providing a useful representation for classification. Previous methods have only used the distances matrices directly, those previous attempts did not allow for interpolation, averaging, classification, or generation. As pointed out, our latent representation also leads to improved performance regarding classification. This is another, additional, contribution.
>
> We also addressed additional benefits of ShapeEmbed in the rebuttal comment for reviewer iSBr (in the section “Added benefit of ShapeEmbed”).
>
>
> [1]: Dokmanic, Ivan, et al. "Euclidean distance matrices: essential theory, algorithms, and applications." IEEE Signal Processing Magazine 32.6 (2015): 12-30.

---

### Official Review · Reviewer_Kna5 · 2025-07-02

**Clarity:** 3
**Significance:** 2
**Originality:** 3
**Rating:** 5
**Confidence:** 3

**Summary:**

This paper introduces ShapeEmbed, a self-supervised framework for learning transformation-invariant 2D shape representations from binary masks. The key idea is to represent an object's shape via its Euclidean distance matrix over uniformly sampled contour points, which is inherently invariant to translation, rotation, and reflection. To achieve scale and contour-index invariance, the authors normalize the matrix and modify a convolutional variational autoencoder (VAE) to include circular padding and reflection-averaged encoding. Furthermore, they define a symmetric reconstruction loss that accounts for all cyclic permutations and mirrored versions of the input, ensuring that the learned latent shape descriptors are fully invariant. The framework is validated across synthetic and real-world datasets, including biological microscopy images, showing strong performance in downstream shape classification and generative applications.

**Questions:**

1. Section 3.2 states that the encoder outputs a shifted latent representation, whereas Section 3.3 claims that the encoder is invariant to indexation. However, a shifted output and indexation invariance are not necessarily equivalent concepts. Could the authors clarify this apparent discrepancy and elaborate on how indexation invariance is formally ensured?

2. The proposed framework seems to rely on high-quality segmentation masks. However, in real-world scenarios, segmentation masks often contain noise. How robust is your method to such noisy inputs? Have you evaluated its sensitivity to imperfect masks?

3. In Section 3.3, you introduce several regularization terms to encourage the output distance matrix to be non-negative, symmetric, and have zero diagonal. However, these soft constraints do not strictly guarantee that the reconstructed matrix satisfies all distance matrix properties. Have you considered constructing a distance matrix that is analytically guaranteed to meet these constraints, possibly through a parameterization rooted in mathematical structure (e.g., analogous to how quaternions can be used to ensure valid rotation matrices, or how eigen-decomposition can yield symmetric matrices)?

**Ethical Concerns:**

["NO or VERY MINOR ethics concerns only"]

**Final Justification:**

The authors have provided clear and satisfactory responses to my main concerns. Their clarification on the encoder’s shift-invariance resolves the confusion around indexation, and the revised explanation is both technically accurate and well-formulated. Their robustness experiments—though only in the supplement—demonstrate practical applicability under noisy masks. I also appreciate their thoughtful discussion on enforcing valid distance matrices and their openness to future improvements. These clarifications strengthen the case for the method’s reliability and broader potential. I am raising my overall score to 5 accordingly.

**Limitations:**

Yes, the authors clearly acknowledge the limitation to simply-connected 2D shapes and suggest potential 3D extensions and nested shape modeling as future work.

**Quality:**

3

**Strengths And Weaknesses:**

**Strengths**.

The paper addresses an important and well-defined problem of learning transformation-invariant shape representations in a self-supervised manner. The key contribution lies in leveraging the distance matrix as an input modality, combined with architectural and loss-based mechanisms to enforce invariance to translation, rotation, scaling, reflection, and contour reindexing. The methodology is elegant, mathematically sound, and requires minimal pre-processing. Compared to prior work like O2VAE and SimCLR-based mask encoders, ShapeEmbed demonstrates superior downstream performance on a range of datasets, especially in challenging biological settings where shape is semantically meaningful.

**Weaknesses**

On the downside, while the paper is thorough in its formulation and evaluation, the scope of the method is limited to simply connected 2D shapes with single contours. It cannot directly handle shapes with holes, disconnected components, or 3D geometries, which limits applicability to more general vision problems. Although experiments cover a reasonable range of datasets and ablations, some aspects—such as the dependence on contour sampling resolution, alternative normalization schemes, or quantitative contour reconstruction fidelity—could be further explored to strengthen the generality and rigor of the evaluation. The reliance on Euclidean distance matrices also may pose computational challenges when scaling to high-resolution contours.

---

> ### Author Rebuttal · Authors · 2025-07-30
>
> ## Limitation of the method to simply-connected 2D shapes with single contours
> The reviewer points out that ShapeEmbed cannot directly handle shapes with holes or disconnected components. Although it is entirely true that shapes with holes are not natively handled by ShapeEmbed, the external outline of an object with holes can be sufficient to describe its shape with no major impact on performance, as is the case with many of the classes in the MNIST dataset (0, 4, 6, 8, and 9). While ShapeEmbed does currently take a single distance matrix as a shape descriptor, it could be modified in the future to allow its input tensor to include additional channels for additional shape components in the same way font glyph descriptors embed several paths to draw letters with holes or disconnected components.
> Regarding 3D geometries, we are currently working on a follow-up model adapted to 3D surfaces, as further elaborated in our response to reviewer pLNK.
>
> ## Shifted output and indexation invariance in ShapeEmbed’s encoder.
> The reviewer asks about the properties of the encoder and writes that “a shifted output and indexation invariance are not necessarily equivalent concepts”. We agree with this. The reviewer’s comment helped us identify that the way we explained the properties of our encoder may cause confusion.
> To elaborate further on this point: choosing a different indexation (i.e. using a different location along the contour as starting point) corresponds to diagonally shifting the input matrix. This effect can be observed in Figure 2 by comparing panels a,b with panels e,f. In Section 2.3, we write that convolutional layers are **equivariant with respect to shifting**, that is, a shifted input (caused, for instance, by choosing a different starting point along the contour) will result in an identical but shifted output. However, the encoder as a whole does not only contain convolutional layers, but also pooling operations. If we now consider a combination of convolutional layers and subsequent global pooling operations, the resulting network will be perfectly **shift invariant**, meaning all shifted versions of an input will result in identical outputs irrespective of how they have been shifted. As we discuss lines 192 to 196, our architecture does not however ensure perfect shift invariance since it uses non-global pooling and strided convolutions. In practice, we find it to provide a sufficient level of invariance as elaborated in Section 4. We propose to modify the following text in Section 4, line 190 as follows to clarify:
>
> *As a result, the convolutional layers are shift-equivariant and produce equal but shifted outputs for all possible distance matrix indexations (starting points). By combining convolutions with a subsequent global pooling operation, we could in principle achieve true shift-invariance. However, ResNet-18 reduces the tensor size in multiple steps using local pooling operations and strided convolutions. Strictly speaking, …*
>
> ## Robustness of the model to noise.
> The reviewer asks for an evaluation of ShapeEmbed’s robustness to noisy segmentation masks. Our Supplementary Material Section G, entitled “Additional Biological Imaging Results”, contains a subsection G3 entitled “Robustness to Segmentation Quality” that is of relevance to address this question. There, we demonstrate ShapeEmbed’s ability to accurately encode shape information even with poor quality, noisy images, resulting in low quality segmentation masks that mimic challenging real-world applications. We synthetically degraded the MEF dataset considered in our manuscript with various levels of image noise, simulating a combination of Poisson noise and Gaussian readout noise as commonly encountered in microscopy image acquisition. Supplementary figure 9 shows examples of images and resulting segmentation masks after degradation with noise. Supplementary table 12 presents a summary of the image statistics of the noise-degraded versions of the MEF dataset. Supplementary Figure 10 reports ShapeEmbed’s performance, measured on the downstream classification task described in the main manuscript (Section 4.2), with the F1-score reported as a metric. These experiments illustrate how noisy data would affect ShapeEmbed’s latent representation quality and are currently referred to in the main manuscript (Section 4.5). Although we could not include these experiments in the main manuscript due to space constraints, we would be open to referencing them more explicitly if deemed useful.
>
> ## Ensuring the validity of predicted distance matrices.
> We very much appreciate the reviewer's suggestion to ensure the prediction of valid distance matrices not via extra loss functions but via analytical guarantees (as outlined for instance in[1]). We see multiple avenues toward achieving this. Possibilities include ideas outlined in[2] , or predicting point coordinates of an arbitrarily shifted and rotated contour, which can then be used to compute the distance matrix to be used in our loss function. While we attempted to implement this early on, we encountered difficulties in training our model with such a hard constraint. In contrast, we empirically observed that training our model with the regularizers we are now using (which are indeed not hard constraints) was comparatively much simpler and sufficient to yield excellent performance. We have continued to explore this area of research and have achieved encouraging results since the submission of our manuscript, but we still consider it to require further exploration and to be beyond the scope of ShapeEmbed as presented here. We would however like to mention this as a possible area of future work in our conclusion section as follows:
>
> *Our current implementation uses several regularization terms in the loss to promote the decoded outputs to be valid Euclidean distance matrices, but lacks strict guarantees that the distance matrix constraints are satisfied. An exciting area for future research could be to instead parameterize the predicted matrix in such a way that distance matrix properties are guaranteed to be valid [1] .*
>
>
> [1]: Dokmanic, Ivan, et al. "Euclidean distance matrices: essential theory, algorithms, and applications." IEEE Signal Processing Magazine 32.6 (2015): 12-30.
>
> [2]: Hoffmann, Moritz, and Frank Noé. "Generating valid Euclidean distance matrices." arXiv preprint arXiv:1910.03131 (2019).

---

> ### Comment · Reviewer_Kna5 · 2025-08-03
>
> The authors have provided clear and satisfactory responses to my main concerns. Their clarification on the encoder’s shift-invariance resolves the confusion around indexation, and the revised explanation is both technically accurate and well-formulated. Their robustness experiments—though only in the supplement—demonstrate practical applicability under noisy masks. I also appreciate their thoughtful discussion on enforcing valid distance matrices and their openness to future improvements. These clarifications strengthen the case for the method’s reliability and broader potential. I am raising my overall score to 5 accordingly.

---

### Note · Authors · 2025-08-11

As we did not receive any further questions from the reviewers, we trust that our responses have fully addressed their concerns.

---

### Decision · Program_Chairs · 2025-09-17

**Decision:**

Accept (poster)

**Comment:**

This paper introduces a self-supervised method for learning transformation-invariant 2D shape representations from binary masks. Reviewers highlighted several strengths: (a) the novelty of combining distance matrices with a latent space learned through a VAE, enforcing invariance to translation, rotation, scaling, reflection, and contour reindexing, and (b) strong experimental results across multiple datasets, particularly in challenging biological settings. Major weaknesses identified were: (a) the method is limited to 2D shapes with single contours and does not currently extend to 3D geometries, (b) the evaluated datasets are somewhat limited, and (c) concerns about scalability and runtime.

The paper received three accepts, one borderline accept, and one borderline reject (VYfD). Four of the five reviewers agreed that the strengths outweighed the weaknesses and that the authors’ responses in the rebuttal and during the author-reviewer discussion adequately addressed their concerns. During the AC-reviewer discussion, the negative reviewer maintained that combining distance matrices and a VAE for self-supervised learning of 2D shape representations “may not meet the novelty threshold” and that “the methodological innovation remains relatively limited.” In contrast, the other reviewers emphasized that the method is novel, noting the absence of prior work encoding contour representations that are invariant to translation, rotation, scaling, reflection, and indexing.

The AC concurs with this latter assessment and further notes that, according to the NeurIPS 2025 reviewer guidelines, originality can be established through “a novel combination of existing techniques,” provided that "the rationale for the combination is well articulated". The AC finds that the submission meets the novelty requirement, and recommends acceptance. The authors are strongly encouraged to integrate the additional experiments, results, and clarifications from the rebuttal and discussion into the final version of the paper!